# Unraveling the cognitive patterns of Large Language Models through module communities

**Kushal Raj Bhandari**
Department of Computer Science
Rensselaer Polytechnic Institute
Troy, NY, USA
bhandk@rpi.edu

**Pin-Yu Chen**
IBM Research
Yorktown Heights,
NY, USA
pin-yu.chen@ibm.com

**Jianxi Gao**
Department of Computer Science
Rensselaer Polytechnic Institute
Troy, NY, USA
gaoj8@rpi.edu

## Abstract

Large Language Models (LLMs) have transformed science, engineering, and society through applications from discovery and diagnostics to chatbots. Yet their mechanisms remain hidden within billions of parameters, making their architecture and cognitive processes difficult to grasp. We address this by drawing on biological cognition and introducing a network-based framework linking cognitive skills, LLM architectures, and datasets. The skill distribution in module communities shows that, while LLMs lack the highly localized specialization of some biological systems, they form distinct module clusters whose skill patterns partly mirror the distributed organization of avian and small mammalian brains. Our results highlight that skill acquisition in LLMs benefits from dynamic, cross-regional interactions and plasticity. Integrating cognitive science and machine learning, this framework offers new interpretability insights and suggests fine-tuning strategies should emphasize distributed learning over rigid modular approaches.

The widespread adoption of LLMs is a testament to their impressive capabilities in generating coherent and context-aware text [3], which has led to their use in everything from customer service chatbots[25] and automated content creation[19] to advanced data analysis[10] and even scientific research [2]. While the practical benefits of LLMs are recognized, a significant gap remains in our understanding of what drives their impressive performance. This imbalance, where the focus is predominantly on leveraging their utility rather than studying their working mechanism, has spurred many questions about the underlying principles that drive their success[2]. Bridging the gap between the widespread use of LLMs and the fundamental principles that drive their performance is a critical challenge. Much like the complexities of the human brain, these systems operate as "black boxes," making it difficult to uncover the mechanisms behind their decision-making.

The complexities of understanding LLMs involve exploring the intriguing parallels and distinctions between artificial neural architectures and the human brain, revealing captivating patterns of resemblance [27, 1] despite their inherent differences [33]. Neuroscientists have long used brain mapping to identify discrete regions with synchronous activity linked to cognitive processes, memory, language, and motor control[14, 30]. We illustrate the distribution of cognitive skills (i.e., cognitive process memory, executive function, language communication, and social cognition) alongside their associated datasets (Figure 1), highlighting the diversity of tasks and the strong alignment between dataset

39th Conference on Neural Information Processing Systems (NeurIPS 2025) Workshop: UniReps: 3rd Edition of the Workshop on Unifying Representations in Neural Models.

categories and core cognitive functions. Moreover, network science approaches have substantially enriched neuroscience by illuminating how large-scale brain networks exhibit modular structures, small-world properties, and dynamic connectivity patterns [50, 55]. Building on such insights, recent studies on LLMs have adopted techniques, employing systematic benchmarks like CogBench[7], psychological tests such as cognitive reflection and semantic illusions[17], and even neuroimaging comparisons to evaluate their cognitive capabilities[4]. Yet, many of these efforts remain unsupported by a cohesive framework rooted in cognitive science and neuroscience, highlighting the importance of systematically mapping the alignment between LLMs and abstract cognitive skills.

Expanding upon this growing interest, recent studies have explored how cognitive skills are encoded and localized within these models. For instance, aligning datasets with linguistic and cognitive skills facilitates targeted training and evaluation of LLM capabilities [5], though such approaches often overlook the role of neural mechanisms that generate these skills. Efforts to map specific tasks onto localized regions of a fine-tuned LLM's architecture [22] reveal the emergence of task-specialized modules, yet they fall short of explaining the structural neural dynamics that support this localization. Similarly, linking in-context learning with cognitive skills has offered insights into the meta-cognitive capabilities of LLMs[22], but these studies primarily focused on behavioral outputs and self-assessment metrics rather than deeper structural explanations. Despite such advancements in understanding cognition in LLMs, prior works lack exploration of inter-skill relationships, dynamic adaptability, cross-domain generalizability, and detailed interpretability of underlying mechanisms.

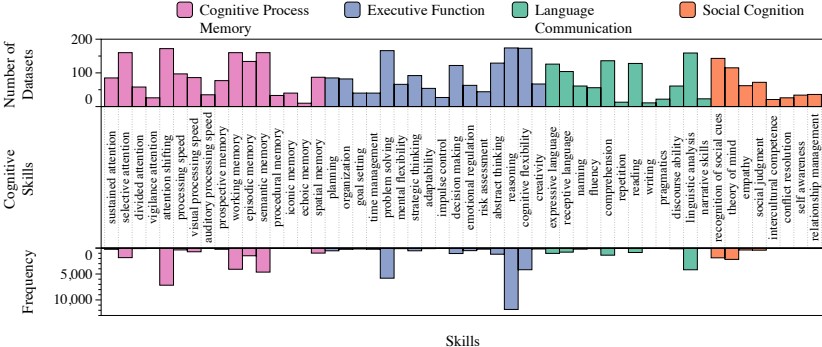

Figure 1: **Categorization and Frequency of Cognitive Skills Across Datasets** The bar plots illustrate how different cognitive functions are categorized by the frequency of their associated skills and how often they appear across multiple-choice question datasets.

## Network Formulation

While cognitive science and neuroscience have long benefited from clearly defined network nodes and edges to uncover the brain's modular organization, where distinct regions support specific cognitive functions [50, 11, 55], establishing an analogous structural topology for LLMs remains a significant challenge. In particular, mapping predefined abstract cognitive skills onto discrete modules within an LLM's architecture, such as attention heads, feedforward blocks, and layer-wise substructures, introduces a novel frontier that challenges conventional approaches to understanding their internal dynamics.

**Skills-Dataset Network, $\mathbf{B}^{\mathbf{DM}}$ (detailed method in ).** To explore such a premise, we employ multi-layered network analysis to examine interconnected LLM modules based on cognitive skills. This analysis examines the complex interactions between cognitive skills, datasets, and modules, providing a detailed perspective on the functional organization within LLMs. As illustrated in Figure 1a., we map different cognitive skills $s_i \in \mathcal{S}$, previously studied in the cognitive science domain (Table S1), to individual multiple-choice problem datasets. Formally, let $s_i \in \mathcal{S}$ denote a set of abstract cognitive skills and $\mathbf{D}_j \in \mathcal{D}$ a collection of multiple-choice question datasets. Each question in dataset $\mathbf{D}_j$ is annotated with a binary skill vector over $\mathcal{S}$ indicating which skills are required. We define the matrix $\mathbf{B}^{\mathbf{SD}} \in \mathbb{R}^{n \times m}$ such that:

$$\mathbf{B}^{\mathbf{SD}}_{ij} = \sum_{k=1}^{r_j} q_i^{(x)}, \tag{1}$$

where $q_i^{(x)} = 1$ if skill $s_i$ is required to solve question $\mathbf{Q}_x$, and $0$ otherwise. Thus, $\mathbf{B}^{\mathbf{SD}}_{ij}$ quantifies the frequency with which skill $s_i$ is required to solve questions within dataset $\mathbf{D}_j$.

This mapping results in a Skill Dataset bipartite network, where cognitive skills, sampled using ChatGPT 3.5, are linked to specific datasets, with the connections weighted by the count of matched skills (detailed in SI 1). The empirical results in Figure 1b show that memory and executive-related skills, such as *reasoning*, *working memory*, *problem-solving*, and *planning*, are well-represented in multiple-choice problems. This highlights the strong alignment of datasets with cognitive functions that lend themselves to structured evaluation. We also observe notable frequencies in other cognitive domains, such as language and communication, and certain aspects of social cognition, reflecting a broader yet still uneven coverage. Importantly, the bar plot on the right of figure 1b underscores the diversity within these datasets, showcasing how they are not uniformly distributed but instead target specific clusters of cognitive functions. This reveals an opportunity to leverage these datasets for analyzing models across a wide range of cognitive abilities. Didolkar et al. provided a similar analysis on utilizing another pre-trained model to generate different abstract cognitive skills for mathematical datasets. Our approach provides more general skills and dataset mapping using existing cognitive science domain literature [22].

**Dataset-Modules Network, $\mathbf{B^{DM}}$ (detailed method in )**: Subsequently, we construct the Datasets vs. Modules network using LLM-Pruner [44], where the modules, defined as subsets of weights, $\mathcal{M}_k \subseteq \mathcal{W}, \forall k \in \{1, 2, \ldots, |\mathcal{M}|\}$, representing structural units of the model (e.g., layers or blocks), are analyzed to assess the impact of datasets on these modules (detailed in SI 2). We quantify the impact of individual multiple-choice question datasets, $\mathbf{D}_j \in \mathcal{D}$, on the individual weight modules of LLM, $\mathcal{M}_k$, using two parameters: change in accuracy after pruning the model to the dataset and fraction of weights pruned within each module. That is, the importance of modules, $\mathbf{B^{DM}}$, is defined as,

$$\mathbf{B}_{jk}^{\mathbf{DM}} = \left(1 - |\Delta\mathrm{acc}(\mathbf{D}_j)|\right) \frac{|\mathcal{M}_k \cap \mathbf{W}_{\mathrm{essential}}|}{|\mathcal{M}_k|} \tag{2}$$

where $\Delta\mathrm{acc}(\mathbf{D}_j)$ denotes the change in accuracy caused by pruning the model with dataset $\mathbf{D}_j$, and $\mathbf{W}_{\mathrm{essential}}$ refers to the set of essential weights identified as critical after pruning the model.

The integration of these two bipartite networks yields a Skills and Modules network, illustrating the relationship between skills and modules and highlighting which modules are influenced by which specific skills. Utilizing equation 1 and 2, we define a projection bipartite network $\mathbf{B^{SM}}$, to project relationship between individual skill $s \in \mathcal{S}_k$ and individual modules $\mathcal{M}_k$,

$$\mathbf{B}_{ik}^{\mathbf{SM}} = \sum_{\mathbf{D}_j \in \mathcal{D}} \mathbf{B}_{ij}^{\mathbf{SD}} \cdot \mathbf{B}_{jk}^{\mathbf{DM}} \tag{3}$$

Further analysis projects these into Modules and Skills networks, revealing the inter-dependencies and collaborative dynamics between modules and the co-dependencies among cognitive skills within the LLM. From equation 3, we describe the relationship between two skills($s_{i_1}$ and $s_{i_2}$) as $\mathbf{P}_{i_1 i_2}^{\mathbf{S}}$, and two modules ($\mathcal{M}_{k_1}$, $\mathcal{M}_{k_2}$) as $\mathbf{P}_{k_1 k_2}^{\mathbf{M}}$.

$$\mathbf{P}_{i_1 i_2}^{\mathbf{S}} = \sum_k^{|\mathcal{M}|} \mathbf{B}_{i_1 k}^{\mathbf{SM}} \cdot \mathbf{B}_{i_2 k}^{\mathbf{SM}}, \quad \text{and} \quad \mathbf{P}_{k_1 k_2}^{\mathbf{M}} = \sum_i^{|\mathcal{S}|} \mathbf{B}_{ik_1}^{\mathbf{SM}} \cdot \mathbf{B}_{ik_2}^{\mathbf{SM}} \quad \text{where,} \ \mathbf{P}^{\mathbf{S}} \in \mathbb{R}^{n \times n} \mathbf{P}^{\mathbf{M}} \in \mathbb{R}^{k \times k}. \tag{4}$$

Expanding on these definitions, the inter-dependencies between skills and modules are quantitatively analyzed to uncover the underlying patterns of association and specialization within the modules. The function $\mathbf{P}^{\mathbf{S}}$ captures the degree of overlap between skills, offering insights into how cognitive skills rely on shared or distinct modules. Similarly, $\mathbf{P}^{\mathbf{M}}$ highlights the co-activation of modules, revealing the extent to which modules work in sync to support various skills. These relationships quantify a metric to identify clusters of skills and modules that exhibit tight integration, shedding light on the modular architecture of LLMs and their alignment with cognitive frameworks. Figure 4 visualizes the Network formulation of the Llama2 model.

## Modular localization characterizes the module network.

Studies in neuroscience have shown that the brain is both modular and functionally specialized. Different regions form tightly connected groups (i.e., modules) that are linked to specific tasks, such as vision, language, memory, and attention [14, 12]. This architecture, observed across species from *C. elegans* to primates, supports the brain processes information efficiently within each module while still communicating across the whole network [18, 13, 32]. Inspired by this, we examine how skills are distributed across different parts of a LLM to better understand their specialized regions and global connectivity of modules through three network metrics of the Modules network: (1)

spectral property of $\mathbf{P^M}$ for understanding the global structural connectivity and robustness; (2) the participation coefficient that quantifies the extent to which individual modules bridge across community boundaries [53]; and (3) the Z-score for the local connectivity [53].

The eigenvalue distributions in Figures 2(a–c) are obtained from the adjacency matrix of the module–module projection network, constructed as described in Equation (4). Figures 2(**a–c**) show the Eigenvalue distribution of the Llama, Llama Chat, and Vicuna models, respectively. All three LLMs consistently show that the modules within communities interact extensively with other communities, indicating that the modules within these networks are tightly knit within communities but loosely connected across different communities. The participation coefficient quantifies the extent of cross-community interactions, while the Z-score captures the relative importance of nodes within their respective communities. Together, these measures provide a more detailed understanding of the roles individual modules play in the community structure of LLM architecture, as shown in Figures 2(**d–f**). The broader distribution of participation coefficient values reflects diverse and well-integrated community dynamics, consistent with the spectral gap, and suggests a network topology that facilitates robust inter-community communication. This pattern is consistent across all three models and is further supported by extended analyses presented in the Supplementary Information(see SI Section 5 and SI Section 6). There, we explore the effects of channel-versus block-level pruning, provide theoretical justification and robustness checks for the observed network properties, and present additional simulations and visualizations that reinforce these findings.

The brain's modular architecture balances functional specialization and global integration, supporting complex yet stable cognitive functions[14]. Our three metrics illustrate comparable patterns of modular localization in LLMs, indicating similar organizational principles emerging across biological and artificial systems [31]. These findings carry significant implications for both fields. In AI, they highlight the potential for designing more efficient and adaptable models by leveraging modularity, mimicking how the brain organizes specialized functions while maintaining flexible interconnectivity. For neuroscience, understanding the extent to which artificial systems replicate biological modularity could inform the study of brain function and network organization, offering insights into how cognitive processes emerge from modular networks.

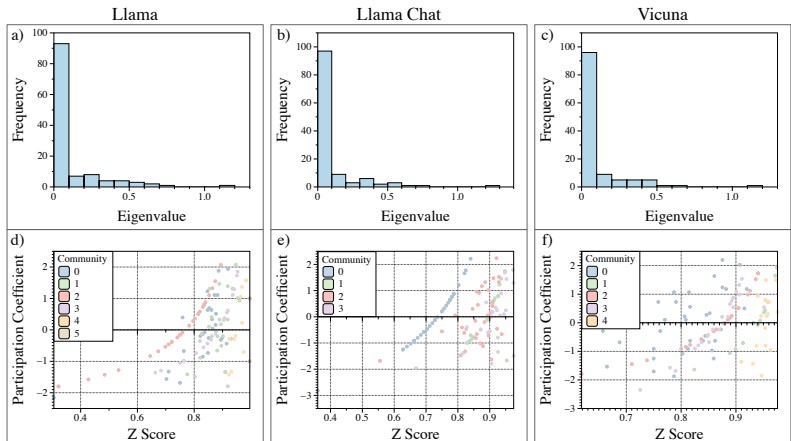

Figure 2: **Spectral property and influence of modules within each community of Modules Network.**(**a–c**) Eigenvalue distributions highlight distinct community structures. (**d–f**) Participation coefficients and Z-scores show bridge modules and identify influential or peripheral nodes.

### Reveal the functional specialization through cognitive Skill-Based Fine-Tuning

To rigorously validate and deepen the impact of our analysis, we must extend our focus beyond network topology and cognitive skills, examining how module communities inform fine-tuning strategies aimed at emulating neural behaviors. In biological systems, we observe three distinct neural architectures: (1) the strong-localization architecture, characterized by isolated subgraphs executing autonomous tasks like octopus [15, 20]; (2) the small-world architecture, which includes a few interconnectivity between communities as seen in the human brain [13, 31]; and (3) the weak-localization architecture, with extensive interconnectivity between communities, typical of avian and small mammalian brains [29]. A key question arises: How can insights from biological functional

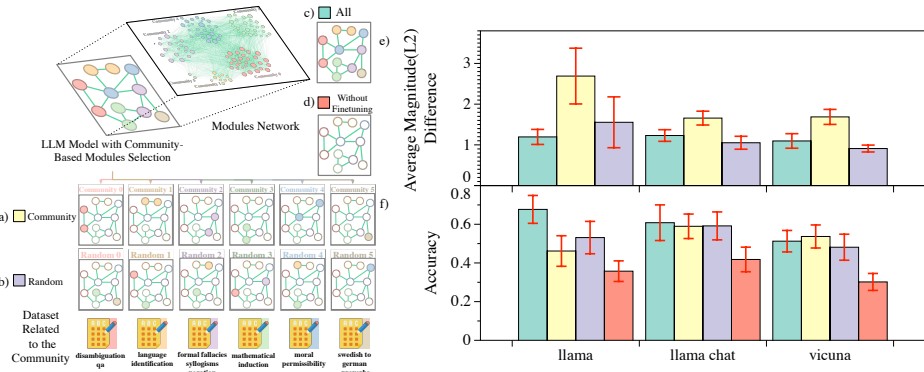

Figure 3: **Comparison of accuracy and magnitude of weight change across fine-tuned LLMs using different cognitive skill-based finetuning. (a–d)** Schematic of fine-tuning approaches: Community (modules aligned to skills), Random (size-matched random subsets), All (all modules), and Without Finetuning (baseline). **(e–f)** Average L2 weight differences and corresponding accuracies using block pruning across community-aligned datasets, highlighting performance differences across fine-tuning strategies.

specialization help explain how cognitive-skill-based communities in LLMs support targeted learning and affect model performance?

To investigate, we fine-tuned models under four configurations (Figure 3 a–d). Learning strengthens synaptic connections via Hebbian learning[8] and long-term potentiation[26], improving neural communication and supporting memory and skill acquisition. Similarly, weight changes after fine-tuning reveal that community-based fine-tuning induces the most substantial adjustments, whereas all-module and random fine-tuning exhibit comparable but lower sensitivity (Figure 3**e**). However, fine-tuning across all modules yields the highest overall accuracy among all configurations (Figure 3**f**), indicating a distinction between distributed knowledge representation in LLMs and the highly localized organization observed in the human brain. However, it aligns with prior findings that task-relevant knowledge in LLMs is redundantly encoded across multiple attention heads in Transformer models [6, 21]. The discrepancy between the extent of structural modifications and the resulting accuracy gains suggests that, although targeting the modules associated with specific cognitive skills induces pronounced parameter changes, these changes do not confer a clear performance advantage. While learning-induced neural plasticity in the human brain is task-specific and efficiency-driven – minimizing disruption to unrelated cognitive functions [11] – community-based fine-tuning in LLMs does not exhibit explicit modular specialization. This result aligns with characteristics of weak-localization architectures, reflecting the compensatory plasticity and cross-regional adaptation observed in large-scale brain networks [30].

Although pre-trained LLMs encode cognitive skills within module communities, targeted interventions do not result in strict functional specialization, prompting a reevaluation of the relative advantages of the three architectural paradigms. In strong-localization architectures like the octopus nervous system, subgraphs function independently, enabling localized learning but limiting global intelligence due to the lack of inter-module support. Small-world architectures, exemplified by the human brain, support task-specific, efficiency-driven learning while minimizing interference with unrelated cognitive functions. Weak-localization architectures, as seen in avian and small mammalian brains, feature specialized neural modules that process distinct cognitive functions but rely heavily on dynamic, cross-regional integration for intelligent behavior [16, 12]. These biological insights align with our observations in LLMs, suggesting that cognitive capabilities do not necessarily benefit from strictly localized fine-tuning. Instead, in weak-localization-like systems, functionality arises from distributed yet interdependent interactions among modular components, underscoring the importance of network-wide coordination for robust cognitive performance.

## Conclusion

This study advances our understanding of LLMs by moving beyond output-based evaluations to probe their internal mechanisms. By integrating concepts from network science and cognitive science, we demonstrated how cognitive skills, datasets, and model modules interrelate, revealing that LLMs exhibit structured communities of modules whose skill activation patterns echo, but do not replicate, biological systems. Our multipartite network analysis showed that while skill clusters

consistently engage similar modules and community-based fine-tuning induces substantial weight changes, these targeted interventions do not outperform random module selections. This underscores a key observation that LLMs' representations are less plastic and more distributed, adapting broadly even when only subsets of modules are tuned.

## Code and Data Availability

Data files and the Python script have been deposited in `https://github.com/KBhandari11/LLMNeuron`
The finetuned weights of all the models have been uploaded in `https://huggingface.co/KBhandari11/collections`.

## Acknowledgment

We acknowledge the support of the US National Science Foundation under Grant No. 2047488 and by the RPI-IBM Future of Computing Research Collaboration(FCRC).

## References

[1] Khai Loong Aw, Syrielle Montariol, Badr AlKhamissi, Martin Schrimpf, and Antoine Bosselut. Instruction-tuning aligns LLMs to the human brain. In *First Conference on Language Modeling*, 2024. URL `https://openreview.net/forum?id=nXNN0x4wbl`.

[2] Rishi Bommasani, Drew A. Hudson, Ehsan Adeli, Russ Altman, Simran Arora, Sydney von Arx, Michael S. Bernstein, Jeannette Bohg, Antoine Bosselut, Emma Brunskill, Erik Brynjolfsson, Shyamal Buch, Dallas Card, Rodrigo Castellon, Niladri Chatterji, Annie Chen, Kathleen Creel, Jared Quincy Davis, Dora Demszky, Chris Donahue, Moussa Doumbouya, Esin Durmus, Stefano Ermon, John Etchemendy, Kawin Ethayarajh, Li Fei-Fei, Chelsea Finn, Trevor Gale, Lauren Gillespie, Karan Goel, Noah Goodman, Shelby Grossman, Neel Guha, Tatsunori Hashimoto, Peter Henderson, John Hewitt, Daniel E. Ho, Jenny Hong, Kyle Hsu, Jing Huang, Thomas Icard, Saahil Jain, Dan Jurafsky, Pratyusha Kalluri, Siddharth Karamcheti, Geoff Keeling, Fereshte Khani, Omar Khattab, Pang Wei Koh, Mark Krass, Ranjay Krishna, Rohith Kuditipudi, Ananya Kumar, Faisal Ladhak, Mina Lee, Tony Lee, Jure Leskovec, Isabelle Levent, Xiang Lisa Li, Xuechen Li, Tengyu Ma, Ali Malik, Christopher D. Manning, Suvir Mirchandani, Eric Mitchell, Zanele Munyikwa, Suraj Nair, Avanika Narayan, Deepak Narayanan, Ben Newman, Allen Nie, Juan Carlos Niebles, Hamed Nilforoshan, Julian Nyarko, Giray Ogut, Laurel Orr, Isabel Papadimitriou, Joon Sung Park, Chris Piech, Eva Portelance, Christopher Potts, Aditi Raghunathan, Rob Reich, Hongyu Ren, Frieda Rong, Yusuf Roohani, Camilo Ruiz, Jack Ryan, Christopher Ré, Dorsa Sadigh, Shiori Sagawa, Keshav Santhanam, Andy Shih, Krishnan Srinivasan, Alex Tamkin, Rohan Taori, Armin W. Thomas, Florian Tramèr, Rose E. Wang, William Wang, Bohan Wu, Jiajun Wu, Yuhuai Wu, Sang Michael Xie, Michihiro Yasunaga, Jiaxuan You, Matei Zaharia, Michael Zhang, Tianyi Zhang, Xikun Zhang, Yuhui Zhang, Lucia Zheng, Kaitlyn Zhou, and Percy Liang. On the opportunities and risks of foundation models, 2022. URL `https://arxiv.org/abs/2108.07258`.

[3] Tom B. Brown, Benjamin Mann, Nick Ryder, Melanie Subbiah, Jared Kaplan, Prafulla Dhariwal, Arvind Neelakantan, Pranav Shyam, Girish Sastry, Amanda Askell, Sandhini Agarwal, Ariel Herbert-Voss, Gretchen Krueger, Tom Henighan, Rewon Child, Aditya Ramesh, Daniel M. Ziegler, Jeffrey Wu, Clemens Winter, Christopher Hesse, Mark Chen, Eric Sigler, Mateusz Litwin, Scott Gray, Benjamin Chess, Jack Clark, Christopher Berner, Sam McCandlish, Alec Radford, Ilya Sutskever, and Dario Amodei. Language models are few-shot learners. In *Proceedings of the 34th International Conference on Neural Information Processing Systems*, NIPS '20, pages 1877–1901, Red Hook, NY, USA, December 2020. Curran Associates Inc. ISBN 978-1-71382-954-6.

[4] Charlotte Caucheteux and Jean-Rémi King. Brains and algorithms partially converge in natural language processing. *Communications Biology*, 5(1):1–10, February 2022. ISSN 2399-3642. doi: 10.1038/s42003-022-03036-1.

[5] Mayee F Chen, Nicholas Roberts, Kush Bhatia, Jue WANG, Ce Zhang, Frederic Sala, and Christopher Re. Skill-it! a data-driven skills framework for understanding and training language models. In *Thirty-seventh Conference on Neural Information Processing Systems*, 2023. URL `https://openreview.net/forum?id=IoizwO1NLf`.

[6] Kevin Clark, Minh-Thang Luong, Urvashi Khandelwal, Christopher D. Manning, and Quoc V. Le. BAM! Born-Again Multi-Task Networks for Natural Language Understanding. In *Proceedings of the 57th Annual Meeting of the Association for Computational Linguistics*, pages 5931–5937, Florence, Italy, 2019. Association for Computational Linguistics. doi: 10.18653/v1/P19-1595.

[7] Julian Coda-Forno, Marcel Binz, Jane X. Wang, and Eric Schulz. CogBench: A large language model walks into a psychology lab. In *Proceedings of the 41st International Conference on Machine Learning*, volume 235 of *ICML'24*, pages 9076–9108, Vienna, Austria, July 2024. JMLR.org.

[8] Ralf Der and Georg Martius. Novel plasticity rule can explain the development of sensorimotor intelligence. *Proceedings of the National Academy of Sciences*, 112(45):E6224–E6232, November 2015. doi: 10.1073/pnas.1508400112.

[22] Aniket Rajiv Didolkar, Anirudh Goyal, Nan Rosemary Ke, Siyuan Guo, Michal Valko, Timothy P Lillicrap, Danilo Jimenez Rezende, Yoshua Bengio, Michael Curtis Mozer, and Sanjeev Arora. Metacognitive capabilities of LLMs: An exploration in mathematical problem solving. In *The Thirty-eighth Annual Conference on Neural Information Processing Systems*, 2024. URL `https://openreview.net/forum?id=D19UyP4HYk`.

[10] Bosheng Ding, Chengwei Qin, Linlin Liu, Yew Ken Chia, Boyang Li, Shafiq Joty, and Lidong Bing. Is GPT-3 a Good Data Annotator? In *Proceedings of the 61st Annual Meeting of the Association for Computational Linguistics (Volume 1: Long Papers)*, pages 11173–11195. Association for Computational Linguistics, 2023. doi: 10.18653/v1/2023.acl-long.626. URL `https://aclanthology.org/2023.acl-long.626`.

[11] Bogdan Draganski, Christian Gaser, Volker Busch, Gerhard Schuierer, Ulrich Bogdahn, and Arne May. Changes in grey matter induced by training. *Nature*, 427(6972):311–312, January 2004. ISSN 0028-0836, 1476-4687. doi: 10.1038/427311a.

[12] Nathan J. Emery and Nicola S. Clayton. The mentality of crows: Convergent evolution of intelligence in corvids and apes. *Science*, 306(5703):1903–1907, 2004. doi: 10.1126/science. 1098410. URL `https://www.science.org/doi/abs/10.1126/science.1098410`.

[13] Lazaros K. Gallos, Hernán A. Makse, and Mariano Sigman. A small world of weak ties provides optimal global integration of self-similar modules in functional brain networks. *Proceedings of the National Academy of Sciences*, 109(8):2825–2830, February 2012. doi: 10.1073/pnas. 1106612109.

[14] Matthew F. Glasser, Timothy S. Coalson, Emma C. Robinson, Carl D. Hacker, John Harwell, Essa Yacoub, Kamil Ugurbil, Jesper Andersson, Christian F. Beckmann, Mark Jenkinson, Stephen M. Smith, and David C. Van Essen. A multi-modal parcellation of human cerebral cortex. *Nature*, 536(7615):171–178, August 2016. ISSN 0028-0836, 1476-4687. doi: 10.1038/ nature18933.

[15] Frank W. Grasso. The octopus with two brains: How are distributed and central representations integrated in the octopus central nervous system? In Anne-Sophie Darmaillacq, Ludovic Dickel, and Jennifer Mather, editors, *Cephalopod Cognition*, pages 94–122. Cambridge University Press, 1 edition, July 2014. ISBN 978-1-139-05896-4 978-1-107-01556-2. doi: 10.1017/ CBO9781139058964.008.

[16] Onur Güntürkün and Thomas Bugnyar. Cognition without Cortex. *Trends in Cognitive Sciences*, 20(4):291–303, April 2016. ISSN 1364-6613. doi: 10.1016/j.tics.2016.02.001.

[17] Thilo Hagendorff, Sarah Fabi, and Michal Kosinski. Human-like intuitive behavior and reasoning biases emerged in large language models but disappeared in ChatGPT. *Nature Computational Science*, 3(10):833–838, October 2023. ISSN 2662-8457. doi: 10.1038/s43588-023-00527-x.

[18] C C Hilgetag, G A Burns, M A O'Neill, J W Scannell, and M P Young. Anatomical connectivity defines the organization of clusters of cortical areas in the macaque monkey and the cat. *Philosophical Transactions of the Royal Society B: Biological Sciences*, 355(1393):91–110, 2000. ISSN 0962-8436. doi: 10.1098/rstb.2000.0551. URL https://www.ncbi.nlm.nih.gov/pmc/articles/PMC1692723/.

[19] Xudong Hong, Asad Sayeed, Khushboo Mehra, Vera Demberg, and Bernt Schiele. Visual Writing Prompts: Character-Grounded Story Generation with Curated Image Sequences. *Transactions of the Association for Computational Linguistics*, 11:565–581, 2023. ISSN 2307-387X. doi: 10.1162/tacl_a_00553. URL https://direct.mit.edu/tacl/article/doi/10.1162/tacl_a_00553/116468/Visual-Writing-Prompts-Character-Grounded-Story.

[20] G. O. Mackie. A Special Invertebrate: *The Anatomy of the Nervous System of Octopus. Science*, 177(4055):1183–1183, September 1972. ISSN 0036-8075, 1095-9203. doi: 10.1126/science.177.4055.1183.a.

[21] Paul Michel, Omer Levy, and Graham Neubig. Are Sixteen Heads Really Better than One? In *Advances in Neural Information Processing Systems*, volume 32. Curran Associates, Inc., 2019.

[22] Abhishek Panigrahi, Nikunj Saunshi, Haoyu Zhao, and Sanjeev Arora. Task-specific skill localization in fine-tuned language models. In *Proceedings of the 40th International Conference on Machine Learning*, ICML'23. JMLR.org, 2023.

[50] Hae-Jeong Park and Karl Friston. Structural and Functional Brain Networks: From Connections to Cognition. *Science*, 342(6158):1238411, November 2013. ISSN 0036-8075, 1095-9203. doi: 10.1126/science.1238411.

[53] Jonathan D Power, Bradley L Schlaggar, Christina N Lessov-Schlaggar, and Steven E Petersen. Evidence for hubs in human functional brain networks. *Neuron*, 79(4): 10.1016/j.neuron.2013.07.035, August 2013. ISSN 0896-6273. doi: 10.1016/j.neuron.2013.07.035.

[25] Stephen Roller, Emily Dinan, Naman Goyal, Da Ju, Mary Williamson, Yinhan Liu, Jing Xu, Myle Ott, Eric Michael Smith, Y-Lan Boureau, and Jason Weston. Recipes for Building an Open-Domain Chatbot. In *Proceedings of the 16th Conference of the European Chapter of the Association for Computational Linguistics: Main Volume*, pages 300–325. Association for Computational Linguistics, 2021. doi: 10.18653/v1/2021.eacl-main.24. URL https://aclanthology.org/2021.eacl-main.24.

[26] Trine Waage Rygvold, Christoffer Hatlestad-Hall, Torbjørn Elvsåshagen, Torgeir Moberget, and Stein Andersson. Long term potentiation-like neural plasticity and performance-based memory function. *Neurobiology of Learning and Memory*, 196:107696, December 2022. ISSN 1074-7427. doi: 10.1016/j.nlm.2022.107696.

[27] Martin Schrimpf, Idan Asher Blank, Greta Tuckute, Carina Kauf, Eghbal A. Hosseini, Nancy Kanwisher, Joshua B. Tenenbaum, and Evelina Fedorenko. The neural architecture of language: Integrative modeling converges on predictive processing. *Proceedings of the National Academy of Sciences*, 118(45):e2105646118, November 2021. ISSN 0027-8424, 1091-6490. doi: 10.1073/pnas.2105646118.

[55] Caio Seguin, Olaf Sporns, and Andrew Zalesky. Brain network communication: Concepts, models and applications. *Nature Reviews Neuroscience*, 24(9):557–574, September 2023. ISSN 1471-0048. doi: 10.1038/s41583-023-00718-5.

[29] Murray Shanahan. The brain's connective core and its role in animal cognition. *Philosophical Transactions of the Royal Society B: Biological Sciences*, 367(1603):2704–2714, October 2012. ISSN 0962-8436, 1471-2970. doi: 10.1098/rstb.2012.0128.

[30] James M. Shine, Michael Breakspear, Peter T. Bell, Kaylena A. Ehgoetz Martens, Richard Shine, Oluwasanmi Koyejo, Olaf Sporns, and Russell A. Poldrack. Human cognition involves the dynamic integration of neural activity and neuromodulatory systems. *Nature Neuroscience*, 22 (2):289–296, February 2019. ISSN 1097-6256, 1546-1726. doi: 10.1038/s41593-018-0312-0.

REFERENCES

[31] Olaf Sporns and Richard F. Betzel. Modular brain networks. *Annual Review of Psychology*, 67(1):613–640, January 2016. ISSN 0066-4308, 1545-2085. doi: 10.1146/annurev-psych-122414-033634.

[32] Gang Yan, Petra E. Vértes, Emma K. Towlson, Yee Lian Chew, Denise S. Walker, William R. Schafer, and Albert-László Barabási. Network control principles predict neuron function in the Caenorhabditis elegans connectome. *Nature*, 550(7677):519–523, October 2017. ISSN 1476-4687. doi: 10.1038/nature24056.

[33] Yuchen Zhou, Emmy Liu, Graham Neubig, Michael J. Tarr, and Leila Wehbe. Divergences between language models and human brains. In *The Thirty-eighth Annual Conference on Neural Information Processing Systems*, 2024. URL https://openreview.net/forum?id=DpP5F3UfKw.

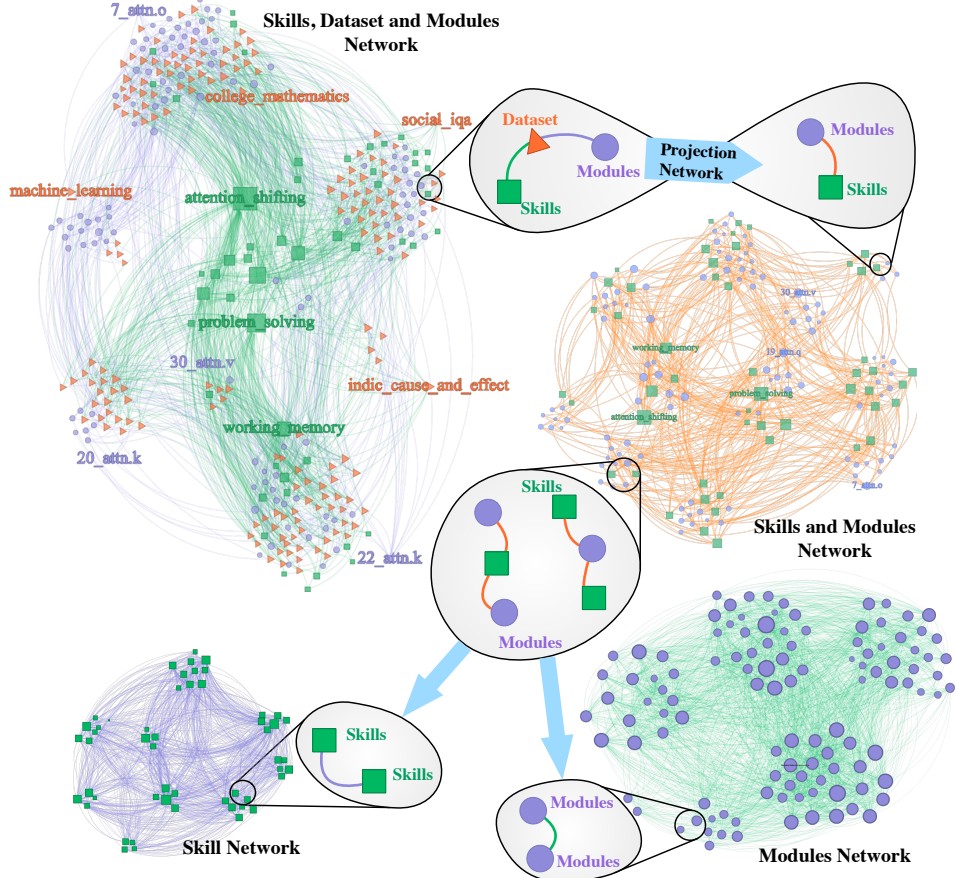

Figure 4: **Multipartite Network of Skills, Dataset and Modules of Llama2 model.** The network depicts modules (squares), datasets (triangles), and skills (circles) as nodes. Edges are weighted by the normalized values derived from the bipartite relationships between skills and datasets and between datasets and modules, reflecting the structural importance and interactions within the multiple types of nodes. The projection network simplifies the multipartite structure by collapsing intermediary nodes(datasets) to focus on the direct interactions between skills and modules using $Block$-based pruning strategy. This projection highlights key dependencies and structural patterns within the model, offering insights into which modules are most influential for specific skills.

## Supplementary Note 1: Skills vs Dataset Network

### Skills

Skills can be conceptualized as abstract cognitive abilities that are essential for solving specific tasks. Formally, we define the set of these abstract skills, denoted by $(\mathcal{S})$, as,

$$\mathcal{S} = \{s_1, s_2, s_3, ..., s_n\}, \tag{5}$$

where $n$ is the total number of skills. Given the inherently abstract nature of $s_i \in \mathcal{S}$, we empirically ground a subset of predefined skills as identified in prior literature. In addition, we also categorize cognitive function as higher-order cognitive skills representing different lower-order cognitive skills ($s_i$) that have been characterized across various domains within cognitive science. Table 1 provides an overview of the subsets of abstract skills considered in this study.

### Dataset

Prior research has been extensively studied to showcase how multiple-choice-based problems can be used to assess different lower- and higher-order cognitive skills [14, 68, 69, 8, 30]. Building on

Table 1: **Cognitive Functions with their corresponding cognitive skills.** The total number of cognitive skills considered is $n = 53$, with each skill $s_i$ categorized into broader higher-order cognitive domains based on classifications from prior literature.

| Category | Cognitive Skills ($\mathcal{S}$) | Citation |
|---|---|---|
| Cognitive Process (Memory) | sustained attention, selective attention, divided attention, vigilance attention, attention shifting, processing speed, visual processing speed, auditory processing speed, prospective memory, working memory, episodic memory, semantic memory, procedural memory, iconic memory, echoic memory, spatial memory | [51, 52, 34, 49, 45, 35, 36, 46, 63, 4, 18, 57, 20, 47] |
| Executive Function | planning, organization, goal setting, time management, problem-solving, mental flexibility, strategic thinking, adaptability, impulse control, decision making, emotional regulation, risk assessment, abstract thinking, reasoning, concept formation, cognitive flexibility, creativity | [39, 41, 17, 54, 1, 27, 64, 28] |
| Language Communication | expressive language, receptive language, naming, fluency, comprehension, repetition, reading, writing, pragmatics, discourse ability, expressive language, receptive language, linguistic analysis, narrative skills | [13, 38, 25, 43, 19, 16, 10] |
| Social Cognition | recognition of social cues, theory of mind, empathy, social judgment, intercultural competence, conflict resolution, self-awareness, relationship management | [6, 21, 2, 7, 48, 23, 26, 5] |

this foundation, we formally define a framework to characterize the relationship between datasets of multiple-choice questions and the cognitive skills required to answer them.

Let $\mathcal{D} = \{\mathbf{D}_1, \mathbf{D}_2, \ldots, \mathbf{D}_m\}$ denote a collection of multiple-choice question datasets, where $m = |\mathcal{D}|$. Each dataset $\mathbf{D}_j \in \mathcal{D}$ contains $r_j$ multiple-choice questions, denoted as $\mathbf{D}_j = \{\mathbf{Q}_1, \mathbf{Q}_2, \ldots, \mathbf{Q}_{r_j}\}$.

Each question $\mathbf{Q}_x \in \mathbf{D}_j$ is associated with a binary skill requirement vector:

$$\mathbf{Q}_x = (q_1, q_2, \ldots, q_n), \tag{6}$$

where $n = |\mathcal{S}|$ and $\mathcal{S}$ is the set of all skills. The value of each component $q_i$ indicates whether skill $s_i \in \mathcal{S}$ is necessary to solve question $\mathbf{Q}_x$:

$$q_i = \begin{cases} 1 & \text{if skill } s_i \text{ is required to answer } \mathbf{Q}_x, \\ 0 & \text{Otherwise.} \end{cases} \tag{7}$$

We then define $\mathbf{B}_{ij}^{\mathbf{SD}}$ as the frequency with which skill $s_i$ appears across all questions in dataset $\mathbf{D}_j$:

$$\mathbf{B}_{ij}^{\mathbf{SD}} = \sum_{x=1}^{r_j} q_i^{(x)}, \qquad \mathbf{B}^{\mathbf{SD}} \in \mathbb{R}^{n \times m} \tag{8}$$

where $q_i^{(x)}$ denotes the $i^{\text{th}}$ component of the skill vector for question $\mathbf{Q}_x$, $n = |\mathcal{S}|$ is the number of distinct skills, and $m = |\mathcal{D}|$ is the number of datasets.

**Mapping Skills to Dataset**

To ensure a comprehensive analysis, we select 174 multiple-choice problem datasets ($m = 174$) spanning a diverse range of domains(MMLU[32], BigBench[60], MathQA[3], CommonsenseQA[62], ScienceQA[42], and TruthfulQA[40]). For each dataset, we select up to $r_{\text{max}}$ questions (or all available questions if the dataset contains fewer than $r_{\text{max}}$) and utilize ChatGPT 3.5 to identify and sample the specific skills required to solve each individual question. That is, for dataset, $\mathbf{D}_j$, number of question we select is

$$r_j = \min\left\{ |\mathbf{D}_j|, r_{\text{max}} \right\}. \tag{9}$$

This approach enables a systematic exploration of the cognitive skills associated with problem-solving across various domains. In this study, $r_{\max} = 100$. Using this sampling approach, we construct the skills dataset bipartite network $B_{ij}^{\mathbf{SD}}$, which represents the number of times different skills $s_i$ are required for solving each dataset within the set of datasets $\mathcal{D}$. This distribution captures the likelihood of each subset of skills $\mathcal{S}$ being required for solving questions in a given dataset.

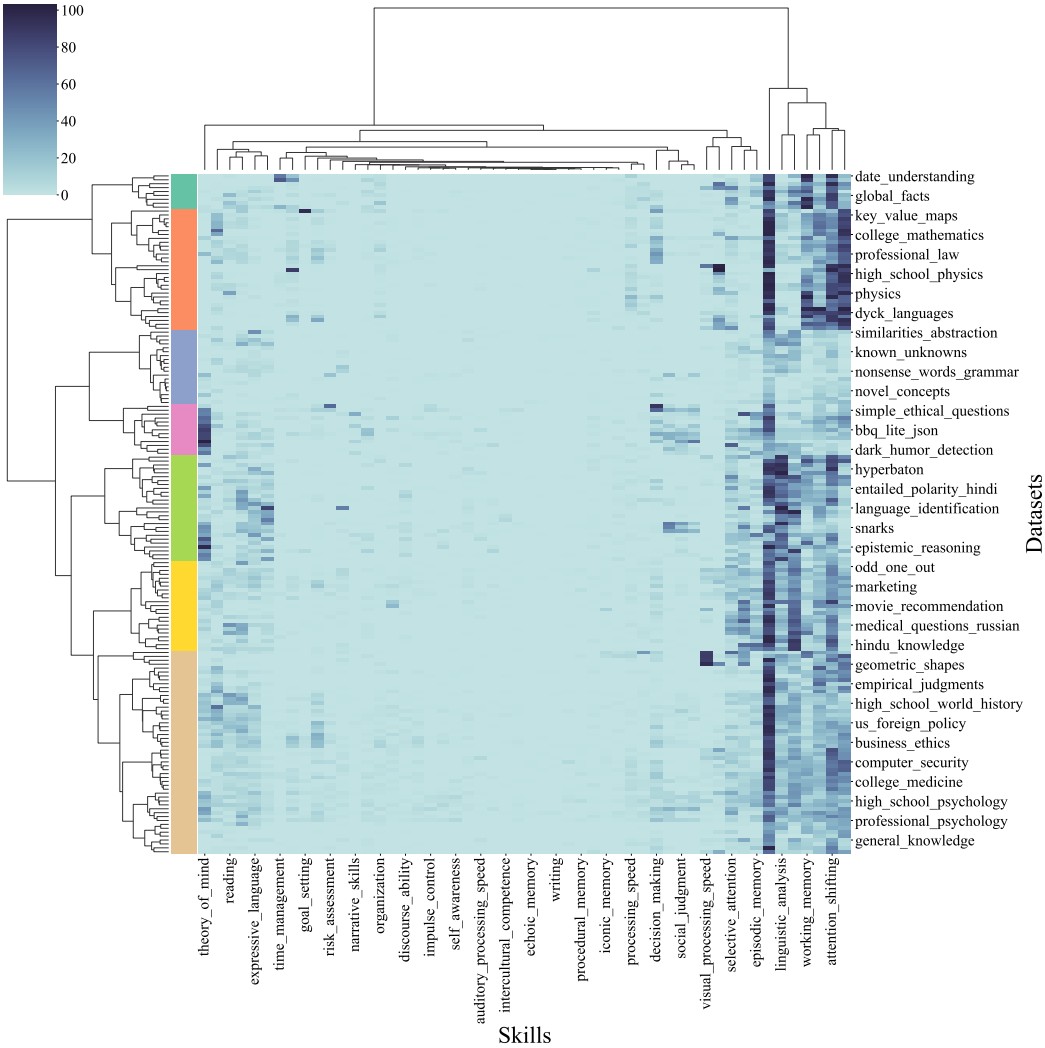

Figure 5: **Heatmap of Skills vs. Datasets ($\mathbf{B}_{ij}^{\mathbf{SD}}$) with Hierarchical Clustering.** The bipartite matrix $\mathbf{B}_{ij}^{\mathbf{SD}}$ represents the number of times skill $s_i$ is required for dataset $\mathbf{D}_j$. Datasets that require similar cognitive skills exhibit strong associations, as indicated by the clustered patterns in the heatmap.

Figure 5, provides better visualization of how different skills are associated with different datasets. To validate the mapping, 8, defined using ChatGPT Prompt utilized for mapping $\mathcal{S}$ to $\mathcal{D}$:

## Supplementary Note 2: Dataset vs Modules

### Pruning

To systematically study how different datasets $\mathbf{D}_j \in \mathcal{D}$ influence the internal modular structure of LLMs, we utilize pruning framework. Prior research has demonstrated that a subset of parameters within a neural network can achieve satisfactory performance [37, 31, 61, 24, 66]. Building on this insight, we prune redundant weight parameters to isolate the critical nodes for each dataset. This

---

**Model:** `gpt-3.5-turbo`
**Message:**

**System:** You are a linguistic and cognitive scientist skilled in analyzing
   texts for their cognitive properties

**Role:** Given different cognitive skills:  {all cognitive skills}
   Select applicable cognitive skills as an unordered list separated
   by commas, which is necessary to answer this question without
   explanation.  {question}

---

Figure 6: **Prompt Template for Cognitive Skill Mapping.** The prompt provides the structure for querying `gpt-3.5-turbo` to identify cognitive skills necessary for answering a specific question. The instruction specifies the context, role, and task, prompting the model to select five relevant cognitive skills from a predefined set of abstract skills.

approach identifies the most activated parameters specific to each dataset. We replicate the pruning strategy across all 174 available datasets, resulting in 174 uniquely pruned models, each tailored to the specific skills required for its respective dataset. Given the large size of LLMs (approximately 7 billion parameters), handling individual parameters directly is computationally prohibitive. Therefore, we focus on individual modules within LLMs. Each module represents a distinct functional component of the model. Our objective is to analyze how the connections between these modules influence the dependencies required for each skill.

**LLM-Pruner**

LLMs possess intricate, modularized architectures, where computation is distributed across various weight matrices, including attention projections (e.g., `attn.q_proj`, `attn.k_proj`, `attn.v_proj`, `attn.o_proj`) and feedforward components (e.g., `mlp.gate_proj`, `mlp.down_proj`, `mlp.up_proj`).

To systematically study the relative importance of these modules in dataset-specific settings, we leverage a networked architecture representation of LLMs by constructing a task-dependent dependency graph, following the LLM-Pruner framework [44]. Ma et al. abstracts LLMs as a directed acyclic graph (DAG), where each node corresponds to an intermediate neuron activation, and each directed edge represents the application of a learnable weight matrix within a specific functional module of the model (e.g., attention projections or feedforward projections). The dependency graph is constructed by statically tracing the model's forward computation, encoding how each activation depends on earlier transformations [44].

We utilize a Taylor expansion-based importance score to rank the significance of each edge for a given dataset. Specifically, the Taylor approximation of the loss change with respect to each edge's removal is computed, providing an efficient estimate of the module's contribution to model performance. We apply both the construction of the dependency graph and the computation of pruning importance, following the original algorithm outlined in [44]. Each individual weight parameter is indexed by $p$, where $p$ identifies a scalar element within the model's collection of weight matrices $\mathcal{W}$ The importance of each individual weight parameter $\mathbf{w}_k$ is computed based on the estimated change in loss $\Delta\mathcal{L}(\mathbf{D}_j)$ for a dataset $\mathbf{D}_j \in \mathcal{D}$, approximated using a Taylor series expansion:

$$I_{\mathbf{w}_p} = |\Delta\mathcal{L}(\mathbf{D}_j)| = \left| \frac{\partial\mathcal{L}(\mathbf{D}_j)}{\partial\mathbf{w}_p}\mathbf{w}_p - \frac{1}{2}(\mathbf{w}_p)^\mathsf{T} H_{pp}\mathbf{w}_p + O(\|\mathbf{w}_p\|^3) \right|, \tag{10}$$

where $H$ denotes the Hessian matrix with respect to $\mathbf{w}_p$.

From equation 10, the importance of weight can be computed efficiently by approximating the Hessian matrix using the Fisher Information method:

$$I_{\mathbf{w}_p} \approx \left| \frac{\partial\mathcal{L}(\mathbf{D}_j)}{\partial\mathbf{w}_p}\mathbf{w}_p - \frac{1}{2}\sum_{k=1}^{r_j}\left( \frac{\partial\mathcal{L}(\mathbf{Q}_x)}{\partial\mathbf{w}_p}\mathbf{w}_p \right)^2 + O(\|\mathbf{w}_p\|^3) \right|, \tag{11}$$

where, $\mathbf{Q}_x$ is an individual question within dataset $\mathbf{D}_j = \{\mathbf{Q}_x\}_{x=1}^{r_j}, \forall\mathbf{D}_j \in \mathcal{D}$.

Using the computed importance scores $I_{\mathbf{w}_p}$ for all parameters, pruning is performed on the dependency graph following two distinct strategies as defined by Ma et al. [44]:

- **Block-Based pruning strategy** targets groups of neurons (blocks) associated with specific functional components, such as an entire attention head or an MLP module. Neurons within these blocks are pruned together, preserving functional coherence while reducing complexity.
- **Channel-Based pruning strategy** systematically removes channels across multiple layers, affecting neurons connected through vertical paths of the network. This method targets entire feature channels, cutting across layer boundaries, and simplifying inter-layer dependencies.

Following pruning, we analyze the induced sparsity across specific architectural modules, including the attention projections and feedforward projections for each transformer layer. Sparsity for each module is computed as the fraction of weights $\mathbf{w}_p$ within that module that have been pruned.

By quantifying the sparsity patterns per dataset $\mathbf{D}_j \in \mathcal{D}$, we capture how different cognitive skill demands $\mathcal{S}$ associated with each dataset map onto the LLM's internal modular structure. This allows a principled exploration of skills-specific modular specialization without modifying the original pruning algorithm of LLM-Pruner.

**Importance of Module**

The importance of the module quantifies the impact of each dataset on individual modules based on the accuracy and sparsity of the modules before and after pruning. Let the individual weight of the LLM be defined as $\mathbf{w}_p \in \mathcal{W}$. Then, modules are subset of weights, $\mathcal{M}_k \subseteq \mathcal{W}$, where $\mathcal{M}_k$ represents all the structural unit (i.e., `attn.q_proj`, `attn.k_proj`, `attn.v_proj`, `attn.o_proj`, `mlp.gate_proj`, `mlp.down_proj`, and `mlp.up_proj`) within all layers of a pretrained model.

From equation 10, weights are filtered based on the sparsity ratio threshold:

$$\mathcal{W}_{\text{pruned}} = \{\mathbf{w}_p \in \mathcal{W} | I_{\mathbf{w}_p} < \tau\} \tag{12}$$

where $\tau$ is a threshold based on the sparsity ratio, and $\mathcal{L}(\mathbf{D}_j)$ is the loss function on dataset $\mathbf{D}_j \in \mathcal{D}$. Thereby, all the essential weights for a particular dataset, $\mathbf{D}_j$, with a $\tau$ sparsity ratio are given by,

$$\mathcal{W}_{\text{essential}} = \mathcal{W} \setminus \mathcal{W}_{\text{pruned}}. \tag{13}$$

The importance of a module is determined by the fraction of its essential weights, those unaffected by pruning, scaled by the complement of the absolute change in accuracy resulting from pruning. This is mathematically expressed as:

$$\mathbf{B}_{jk}^{\mathbf{DM}} = \left(1 - |\Delta\text{acc}(\mathbf{D}_j)|\right) \frac{|\mathcal{M}_k \cap \mathbf{W}_{\text{essential}}|}{|\mathcal{M}_k|} \tag{14}$$

Here, $|\mathcal{M}_k \cap \mathbf{W}_{\text{essential}}|$ represents the count of essential weights in the module, while $|\mathcal{M}_k|$ is the total number of weights in the module. The term $|\Delta\text{acc}(\mathbf{D}_j)|$ measures the absolute change in the model's accuracy caused by pruning. The complement, $1 - |\Delta\text{acc}(\mathbf{D}_j)|$, reflects how much accuracy is preserved, emphasizing the module's robustness. A larger change in accuracy indicates a more adverse effect on performance, reducing the importance of the module. Conversely, a higher sparsity ratio suggests less pruning of the module, thereby increasing its importance. By leveraging the importance of modules, we construct a bipartite network connecting each dataset to all the modules, capturing the relationship between tasks and model components.

Figure 7a, demonstrates the negative correlation between the average $\mathbf{B}_{j,k}^{\mathbf{DM}}$ and the magnitude of $|\Delta\text{acc}(\mathbf{D}_j)|$—the performance drop for the model after pruning. This empirically highlights that datasets with higher $|\Delta\text{acc}(\mathbf{D}_j)|$ (i.e., larger drops in accuracy) are associated with lower $\mathbf{B}_{j,k}^{\mathbf{DM}}$, indicating that modules influence for such datasets are less essential. Such discussion pertains to how much contribution the module makes to the overall performance of solving the task within dataset $\mathbf{D}_j$. If the performance decrease is sharp, then regardless of how significant the pruning, the module's contribution is significantly less to quantify the importance of the modules.

In figure 7(**b** and **c**), the distributions of the sparsity ratio for individual modules and $\mathbf{B}_{j,k}^{\mathbf{DM}}$ in the Llama2 model with $25\%$ pruning illustrate the differences between two pruning strategies. These

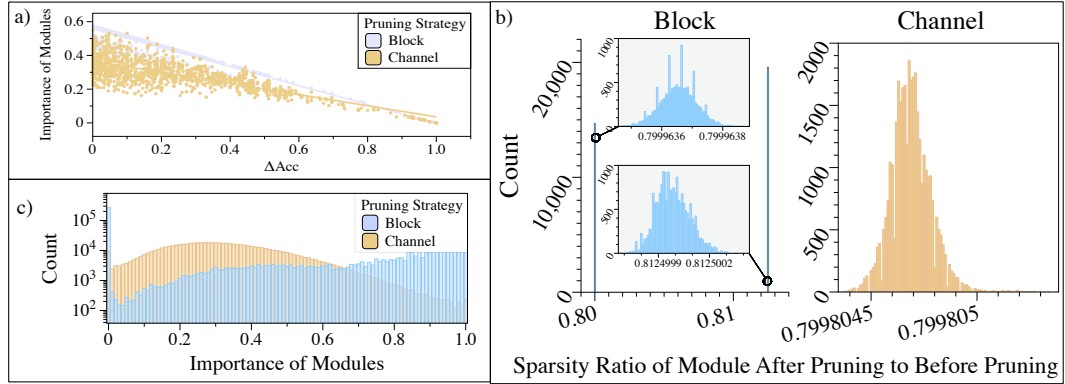

Figure 7: **Average Performance of the pruned model after pruning with different sparsity ratios.** (**a**) Importance of Modules quantifies the relationship between each dataset and the weight modules of LLMs (`Llama2` with a $25\%$ pruning ratio). The scatter plot with the line of best fit shows the relationship between the average $\mathbf{B}^{\mathbf{DM}}_{:k}$ of all LLM modules($\mathcal{M}_k$) and the change in overall model performance before and after pruning. (**b**) The module sparsity ratio distribution of `Llama2` with a $25\%$ pruning ratio is shown for two pruning strategies.(**c**) The variation of $\mathbf{B}^{\mathbf{DM}}$ (edge weight between datasets and individual LLM modules) is shown for two pruning strategies.

differences arise from the inherent structural characteristics of each strategy. The dependency graph for weight parameters using the block-based pruning strategy is independent of the modules, meaning each dependency graph distinguishes different modules, like attention-based modules, from MLP-based modules more distinctly, resulting in a bimodal distribution for the sparsity ratio and $\mathbf{B}^{\mathbf{DM}}_{j,k}$. In contrast, the channel-based pruning strategy leads to a Gaussian distribution, as its dependency graph is more interconnected across all modules. This finding emphasizes how the structural pruning process reveals the sensitivity of model performance to specific datasets and their associated modules, offering a framework to assess dataset dependencies and module importance.

**Compare Activation Pattern with Dataset association**

Foremost, we focus on empirically verifying that the gradient-based structural pruning method applied to LLMs is a valid method for studying the influence of each dataset on the modules. Gradient-based pruning can selectively deactivate neural network weights by identifying and pruning those that contribute the least to the model's performance on specific datasets using weights that activate the least for the dataset [37, 31, 61, 24, 66, 44]. This method generates distinct activation pathways for different datasets, effectively separating module activation based on input characteristics, i.e., skills required to solve the multiple-choice problem. We utilize LLM-Pruner, a state-of-the-art pruning method that utilizes structural and gradient-based methods for large language models (detailed in SI 2).

The influence of each dataset on individual modules is quantitatively assessed using sparsity values, which serve as a metric to gauge the extent of impact. The pruning method inversely exhibits the effect of datasets on the modules, meaning that higher sparsity values indicate a more significant effect of the dataset on the respective modules. Given the high dimensions of sparsity values across all modules and datasets, we utilize Principal Component Analysis (PCA) to comprehensively reduce the dimensions to represent the sparsity patterns. Following PCA, K-Means clustering is applied to the reduced data to identify and group similar patterns. Figure 8 (**a**), represents the optimal number of K-Means clusters that separate different datasets based on their sparsity value of the modules. Figure 8 (**b**), visualizes the scatter plot of different datasets differentiated by the optimal clusters. In addition, we include random structural pruning to highlight the difference between gradient-based pruning using all the datasets. This clustering process highlights how different groups or clusters are characterized and distinguished based on the underlying sparsity patterns, providing insights into the variation in dataset impact across modules.

We further analyze the effectiveness of the pruning approach using Hotelling's T-squared statistic, comparing PCA values of each cluster, including the randomly pruned models. Figure 8(**c,d**) presents

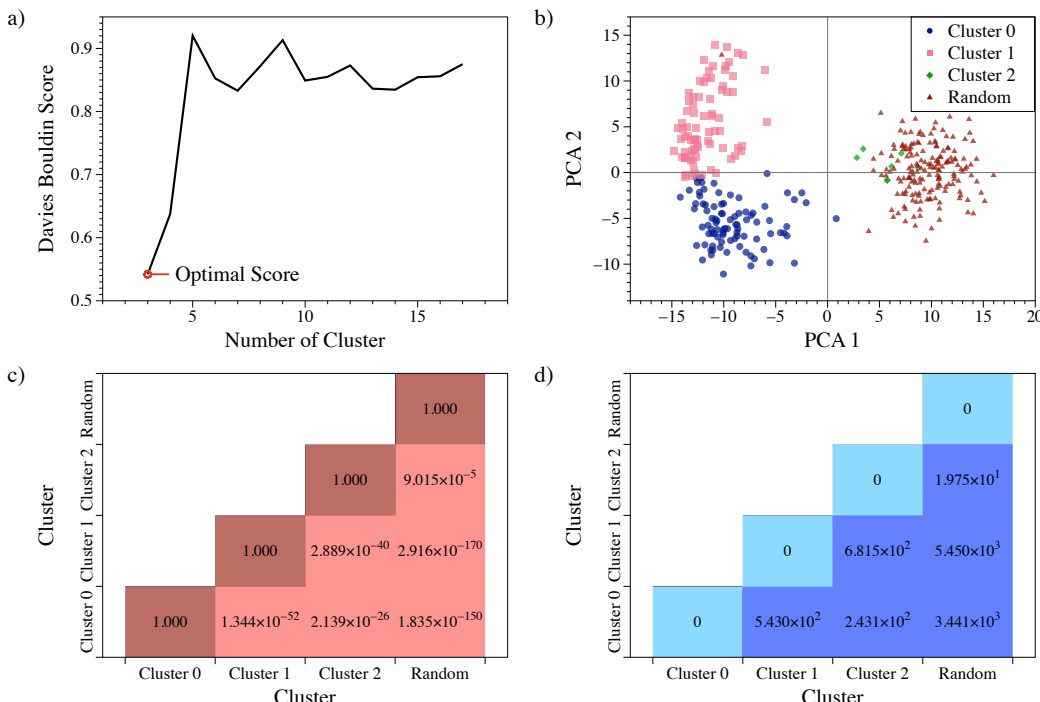

Figure 8: **Evaluating clustering of pruned Llama models using Davies-Bouldin Score and Hotelling's T-squared test.(a)** The Davies-Bouldin Score determines the optimal number of K-Means clusters for grouping 174 PCA values derived from module sparsity values, obtained by pruning the Llama2 model using 174 different datasets. **(b)** A scatter plot showing the pruned Llama2 model, grouped by the optimal number of clusters identified in **a**, alongside randomly pruned models with the 174 datasets. **(c)** P-values from Hotelling's T-squared test between different clusters, including random pruning, are all significantly small ($< 0.05$), indicating dissimilarity in information processing between different clusters. **(d)** Hotelling's T-squared statistics highlight the differences between clusters.

the p-values and statistical results obtained using Hotelling's T-squared test, offering robust evidence that different modules of sparsity value of different modules. The results with a p-value below 0.001 indicate a statistically significant distinction between the clusters produced by the pruning method, including those formed through random pruning. This notable difference implies that the pruning method successfully captures and retains the LLM's information processing characteristics, which are specific to different datasets.

## Supplementary Note 3: Skill Weight Function

Utilizing equation 8 and 14, we define a projected bipartite network, $\mathbf{B}_{ik}^{\mathbf{SM}}$, which quantifies the relationship between module $\mathcal{M}_k$ and the skill $s_i$. This network projects the skill dataset bipartite network $\mathbf{B}_{ij}^{\mathbf{SD}}$ of a dataset $\mathbf{D}_j \in \mathcal{D}$ with the importance of modules $\mathbf{B}_{jk}^{\mathbf{DM}}$, providing a unified measure of module relevance for skills. Mathematically, it is expressed as:

$$\mathbf{B}_{ik}^{\mathbf{SM}} = \sum_{\mathbf{D}_j \in \mathcal{D}} \mathbf{B}_{ij}^{\mathbf{SD}} \cdot \mathbf{B}_{jk}^{\mathbf{DM}} \tag{15}$$

where $\mathbf{B}_{ij}^{\mathbf{SD}}$ represents the skills, $s_i \in \mathcal{S}$, required to solve questions within the dataset $\mathbf{D}_j \in \mathcal{D}$, and $\mathbf{B}_{jk}^{\mathbf{DM}}$ measures the importance of module $\mathcal{M}_k$ based on the fraction of its essential weights scaled by the complement of the accuracy drop caused by pruning the dataset $\mathbf{D}_j \in \mathcal{D}$. This formulation enables targeted analysis of module relevance for specific skills and datasets, offering insights into skill-specific module contributions, dataset selection, and pruning strategies. The projected bipartite

network bridges the gap between skill requirements and model architecture, facilitating informed decisions in model optimization.

In addition, the bipartite network($\mathbf{B}^{DM}$ and $\mathbf{B}^{SM}$) connects skills to datasets and modules, with projections providing a detailed view of the interdependencies. However, summing two different bipartite networks results in a significantly dense network. Projecting this dense network would further amplify its density. To address this, we employ spectral sparsification [59] to reduce the network's density while preserving the largest eigenvalue, thereby maintaining the spectral topology of the original network. Given the stochastic nature of spectral sparsification, the resultant networks vary across different iterations. Skills and modules frequently interacting through common datasets form a projection network, indicating shared functionality or reliance on overlapping cognitive processes.

## Supplementary Note 4: Skills Connectivity Network

From equation 15, we define a projected relationship between two skills, $s_{i_1}$ and $s_{i_2}$, using the metric $\mathbf{P}^{\mathbf{S}}_{i_1 i_2}$:

$$\mathbf{P}^{\mathbf{S}}_{i_1 i_2} = \sum_k^{|\mathcal{M}|} \mathbf{B}^{\mathbf{SM}}_{i_1 k} \cdot \mathbf{B}^{\mathbf{SM}}_{i_2 k}. \qquad \text{where, } \mathbf{P}^{\mathbf{S}} \in \mathbb{R}^{n \times n} \tag{16}$$

The dependency, $\mathbf{P}^{\mathbf{S}}_{i_1 i_2}$, aggregates the product of the associations of each skill with individual modules, reflecting how frequently two distinct cognitive skills activate the same underlying modules within the model's architecture. A high value of $\mathbf{P}^{\mathbf{S}}_{i_1 i_2}$ indicates that the underlying computations required for both skills are not independent but instead share representational resources. Conversely, a lower value implies that the skills are interdependent and likely utilize standard cognitive processes within the LLM.

To assess how closely the empirically detected communities of skills align with cognitive functions as defined in Section , we use the Adjusted Rand Score (ARS) as a robust clustering comparison metric. Specifically, we compute ARS values between the skill communities obtained via Louvain community detection and the predefined cognitive-function labels across different sparsity levels used in pruning the model [65, 33, 15]. The ARS extends the Rand Index(RI)[33] by correcting for chance agreement, providing a normalized measure that accounts for the expected similarity of two random partitions. This makes it particularly useful when comparing communities of cognitive skills of different sizes, since the number of clusters is not fixed.

The Rand Index, RI, measures the proportion of agreement between two clusters by evaluating all pairs of elements and counting how many are assigned together or separately in both partitions. It is defined as:

$$\text{RI} = \frac{a + b}{\binom{n}{2}}, \tag{17}$$

where $a$ is the number of pairs of elements that are in the same cluster in both partitions, $b$ is the number of pairs that are in different clusters in both partitions, and $\binom{n}{2}$ is the total number of possible pairs. The Adjusted Rand Score corrects this index for chance. Formally,

$$\text{ARS} = \frac{\text{RI} - \mathbb{E}[\text{RI}]}{\max(\text{RI}) - \mathbb{E}[\text{RI}]}, \tag{18}$$

where $\mathbb{E}[\text{RI}]$ is RI's expected value under random labeling, and $\max(\text{RI})$ is the maximum possible value of the index. The ARS ranges from -1 to 1, with 1 indicating perfect alignment, 0 suggesting random alignment, and negative values indicating less alignment than expected by chance.

Similarly, we also evaluate the agreement between the skill communities and the cognitive function with the Adjusted Normalized Mutual Information (Adjusted NMI), an information–theoretic metric that quantifies the reduction in uncertainty about one partition given knowledge of the other while correcting for chance overlap [65].

The (unnormalized) mutual information between the two partitions(U and V) is

$$\text{MI}(U, V) = \sum_{c \in U} \sum_{\ell \in V} \frac{N_{c\ell}}{N} \log\left(\frac{N_{c\ell} N}{N_c N_\ell}\right), \tag{19}$$

where $N$ is the number of skills, $N_c$ and $N_\ell$ denote the sizes of cluster $c$ and label class $\ell$, and $N_{c\ell}$ counts skills common to both. Mutual information is normalized to $[0, 1]$ by

$$\text{NMI}(U, V) \;=\; \frac{2\,\text{MI}(U, V)}{H(U) + H(V)}, \quad H(U) \;=\; -\sum_{c \in U} \frac{N_c}{N} \log\!\left(\frac{N_c}{N}\right), \tag{20}$$

yet this value remains biased upward when partitions coincide merely by chance. The adjusted form removes such bias:

$$\text{Adjusted NMI} \;=\; \frac{\text{MI}(U, V) - \mathbb{E}[\text{MI}]}{\max\{H(U),\, H(V)\} - \mathbb{E}[\text{MI}]}, \tag{21}$$

where $\mathbb{E}[\text{MI}]$ is the expected mutual information under random labelings. Adjusted NMI equals 1 for identical partitions, approaches 0 when alignment is no better than chance, and can be negative for non-correlated assignments.

We further utilize the Jaccard Similarity Index, which focuses exclusively on the reproducibility of positive co-assignments. The Jaccard Similarity between the two partitions (U and V) is defined as

$$\text{Jaccard Similarity}(U, V) \;=\; \frac{|U \cap V|}{|U \cup V|}. \tag{22}$$

By applying this metric, we quantitatively evaluate how well the modular structure inferred from the projection matrix $\mathbf{P}^{\text{S}}$ aligns with functional cognitive function.

Figure 9 reveals that alignment between the community of skills and cognitive function defined in remains weak across multiple different pruning strategies. For both block-based (**a–c**) and channel-based (**d–f**) strategies, the adjusted NMI, ARS, and Jaccard Similarity cluster around the level ($\approx 0$) for every sparsity ratio and all three models. Adjusted NMI values oscillate between roughly $-0.05$ and $0.10$, ARS between $-0.05$ and $0.08$, and the Jaccard Similarity score never exceeds $0.15$. We find that for any pruning strategy or ratio, it either leaves the scores unchanged or causes minor fluctuations. Because adjusted NMI and ARS are adjusted for chance, these near-zero trajectories indicate that the skill communities are, at best, only as informative as a random partition. The consistently low Jaccard Index reinforces this conclusion, showing that very few skill pairs classified together by the model correspond to the same cognitive functions.

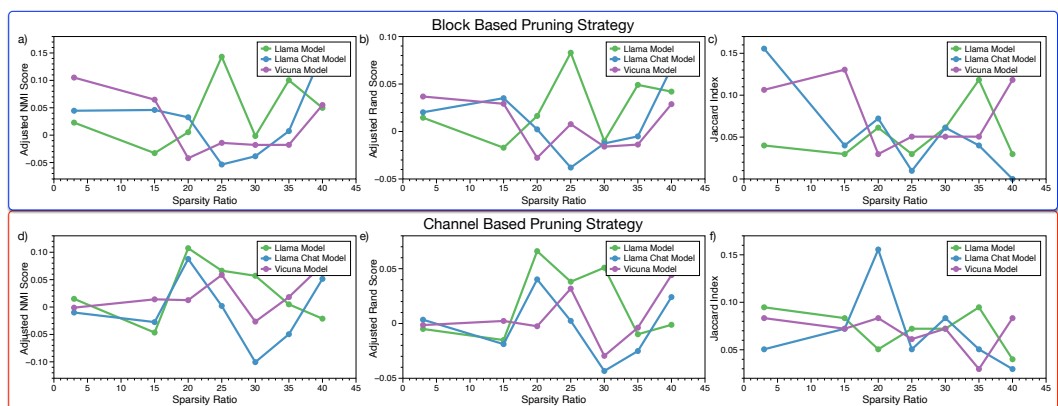

Figure 9: **Comparison of community alignment measures between communities in the Skills network and ground-truth cognitive-function labels, across different sparsity levels.** Subplots (**a–c**) show results for the block-based pruning strategy, while (**d–f**) display the same for channel-based pruning. Each row visualizes the trends for three base models (Llama, Llama-Chat, and Vicuna), using (**a, d**) Adjusted Normalized Mutual Information (NMI), (**b, e**) Adjusted Rand Index (ARI), and (**c, f**) Jaccard Index as similarity metrics. The x-axis denotes the sparsity ratio applied during pruning, enabling evaluation of how sparsity for pruning the LLMs impacts community alignment within cognitive function labels.

## Supplementary Note 4: Modules Connectivity Network

Similarly, to extend the bipartite network $\mathbf{B}_{ik}^{\mathbf{SM}}$ and establish a relationship between two modules, $\mathcal{M}_{k_1}$ and $\mathcal{M}_{k_2}$, we project the skill-based importance to measure their connectivity. This projection is formulated by summing over all skills in $\mathcal{S}$, capturing the shared importance of both modules across the skill space. The connection strength between modules is given by:

$$\mathbf{P}^{\mathbf{M}}_{k_1 k_2} = \sum_{i}^{|\mathcal{S}|} \mathbf{B}_{ik_1}^{\mathbf{SM}} \cdot \mathbf{B}_{ik_2}^{\mathbf{SM}}. \qquad \text{where,} \ \mathbf{P}^{\mathbf{M}} \in \mathbb{R}^{k \times k}, \tag{23}$$

where $\mathbf{B}_{ik_1}^{\mathbf{SM}}$ and $\mathbf{B}_{ik_2}^{\mathbf{SM}}$ represent the bipartite skills modules network $\mathcal{M}_{k_1}$ and $\mathcal{M}_{k_2}$ influenced, respectively, for skill $s_i \in \mathcal{S}$. This projection emphasizes modules relevant to overlapping skills, effectively creating a skill-informed connectivity measure. The resulting metric can be interpreted as the degree of alignment or complementarity between modules in addressing the same skill requirements.

This approach enables the construction of a network of modules, where edges between modules are weighted based on their shared skill-based connectivity. Such a projection allows for a detailed analysis of inter-module interactions. It facilitates the identification of communities within the network, revealing clusters of modules that collectively contribute to specific skill sets. This community detection can further inform optimization strategies by highlighting interdependency and structural relationships within the model architecture, enabling targeted enhancements or pruning.

### Spectral Analysis of Module Connectivity

To analyze the structural properties of the module connectivity network, we utilize the projection network matrix, $\mathbf{P}^{\mathbf{M}}$.

The matrix $\mathbf{P}^{\mathbf{M}}$ is positive semi-definite, we verify that for any non-zero vector $\mathbf{x} \in \mathbb{R}^k$, the following condition holds:
$$\mathbf{x}^{\top} \mathbf{P}^{\mathbf{M}} \mathbf{x} \geq 0.$$
We know,

by substituting $\mathbf{P}^{\mathbf{M}}$ with $\mathbf{B}^{\mathbf{SM}} \mathbf{B}^{\mathbf{SM}}$, since the module connectivity network is a projection network of skills modules bipartite network, 23, we have:
$$\mathbf{x}^{\top} \mathbf{P}^{\mathbf{M}} \mathbf{x} = \mathbf{x}^{\top} (\mathbf{B}^{\mathbf{SM}} \mathbf{B}^{\mathbf{SM}^{\top}}) \mathbf{x} = (\mathbf{B}^{\mathbf{SM}^{\top}} \mathbf{x})^{\top} (\mathbf{B}^{\mathbf{SM}^{\top}} \mathbf{x}).$$
The term $(\mathbf{B}^{\mathbf{SM}^{\top}} \mathbf{x})^{\top} (\mathbf{B}^{\mathbf{SM}^{\top}} \mathbf{x})$ represents the Euclidean norm squared of the vector $\mathbf{B}^{\mathbf{SM}^{\top}} \mathbf{x}$:
$$(\mathbf{B}^{\mathbf{SM}^{\top}} \mathbf{x})^{\top} (\mathbf{B}^{\mathbf{SM}^{\top}} \mathbf{x}) = \|\mathbf{B}^{\mathbf{SM}^{\top}} \mathbf{x}\|^2.$$
Since the squared Euclidean norm of any vector is always non-negative, it follows that:
$$\|\mathbf{B}^{\mathbf{SM}^{\top}} \mathbf{x}\|^2 \geq 0.$$
Hence, $\mathbf{x}^{\top} \mathbf{P}^{\mathbf{M}} \mathbf{x} \geq 0 \quad$ for all $\mathbf{x} \in \mathbb{R}^m$.

Thus, the matrix $\mathbf{P}^{\mathbf{M}}$ is positive semi-definite. Since $\mathbf{P}^{\mathbf{M}}$ is symmetric and positive semi-definite, it can be decomposed using its spectral decomposition:
$$\mathbf{P}^{\mathbf{M}} = \mathbf{U} \mathbf{\Lambda} \mathbf{U}^{\top},$$
where:

- $\mathbf{U}$ is an orthogonal matrix ($\mathbf{U}^{\top} \mathbf{U} = \mathbf{I}$), whose columns are the eigenvectors of $\mathbf{P}^{\mathbf{M}}$.
- $\mathbf{\Lambda}$ is a diagonal matrix containing the eigenvalues of $\mathbf{P}^{\mathbf{M}}$, denoted as $\lambda_1, \lambda_2, \ldots, \lambda_m$.

Since $\mathbf{P}^{\mathbf{M}}$ is positive semi-definite, all eigenvalues $\lambda_i \geq 0$. The rank of $\mathbf{P}^{\mathbf{M}}$ equals the rank of $\mathbf{B}^{\mathbf{SM}}$, implying that the number of non-zero eigenvalues corresponds to the linearly independent columns of $\mathbf{B}^{\mathbf{SM}}$. Eigenvectors associated with larger eigenvalues capture directions in the module connectivity space that reflect dominant patterns of variance, with the largest eigenvalue $\lambda_{\max}$ indicating the most significant connectivity pattern. Conversely, eigenvalues close to zero represent negligible or orthogonal contributions to the connectivity structure. Spectral properties of $\mathbf{P}^{\mathbf{M}}$ can be analyzed to infer community structures: clusters in the connectivity network correspond to large eigenvalues, with coherent eigenvector components highlighting interconnected groups of modules[58].

**Relationship Between $\mathbf{B}^{SD}$, $I_{\mathbf{w}_p}$, and Community Formation**

The skill mapping $\mathbf{B}^{SD}$ acts as a weighting factor for $\mathbf{P}^{\mathbf{M}}$. $\mathbf{P}^{\mathbf{M}}$ emphasizes connections between modules that contribute to the subsets of skills that activate together when solving the task in datasets $\mathbf{D}_j$ that require overlapping skills, Conversely, when $\mathbf{B}^{\mathbf{SD}}$ is highly diverse across datasets, $\mathbf{P}^{\mathbf{M}}$ exhibits weaker block structures.

Similarly, the gradient-based importance measure, $I_{\mathbf{w}_p}$, affects $\mathbf{P}^{\mathbf{M}}$ via its influence on $\mathbf{B}^{\mathbf{DM}}$:

$$\mathbf{B}^{\mathbf{DM}} \propto \frac{\text{Essential Weights in } \mathcal{M}_k}{\text{Total Weights in } \mathcal{M}_k}. \tag{24}$$

Large $I_{\mathbf{w}_p}$ values indicate critical weights that enhance $\mathbf{B}^{\mathbf{DM}}$, creating strong module-skill connections and increasing community cohesiveness.

Combining $I_{\mathbf{w}_p}$ and $\mathbf{B}^{\mathbf{SD}}$, the matrix $\mathbf{P}^{\mathbf{M}}$ encodes community structures that balance:

- **Skill Association (via $\mathbf{B}^{\mathbf{SD}}$):** Modules with similar skill profiles are more likely to cluster together.
- **Weight Importance (via $I_{\mathbf{w}_p}$):** Essential weights amplify module importance, creating stronger module-skill connections.

For a given skill subset $\mathcal{S}_g \subseteq \mathcal{S}$, the contribution of a dataset $\mathbf{D}_j$ to module community formation is proportional to:

$$\mathbf{P}^{\mathbf{M}}_{ij} \propto \sum_{\mathbf{D}_j \in \mathcal{D}} \left[ \frac{I^i_{\mathbf{w}_p} \cdot I^j_{\mathbf{w}_p}}{\|\mathbf{w}_p\|} \right] \mathbf{B}^{SD^2}, \tag{25}$$

The dense blocks in $\mathbf{P}^{\mathbf{M}}$ emerge when modules share overlapping skills and retain essential weights ($I_{\mathbf{w}_p}$), highlighting the importance of both skill association and weight importance. The diversity or concentration of $\mathbf{B}^{\mathbf{SD}}$ dictates the sharpness of community boundaries, while pruning affects the structure by potentially weakening connections for aggressive thresholds (high $\tau$). Balanced pruning, however, preserves meaningful differentiation, enabling $\mathbf{P}^{\mathbf{M}}$ to effectively bridge module importance and skill association, driving community formation in weight and skill spaces.

**Community Detection within Modules Network**

In this study, we employ a robust community detection approach leveraging the Louvain algorithm[9], followed by hierarchical clustering[67], to enhance the stability and reliability of the detected communities. The methodology consists of running the Louvain community detection algorithm 100 times on the same network to capture different possible community structures due to the stochastic nature of the algorithm. Using the results from these multiple runs, a co-assignment matrix is constructed to quantify the frequency with which pairs of nodes are assigned to the same community across different iterations. This co-assignment matrix is then processed using hierarchical clustering with Ward's linkage method to identify clusters of nodes based on their co-assignment frequencies. The final number of communities is determined by selecting the maximum cluster count from the hierarchical clustering results, representing the final community structure of the network. This multi-step procedure improves the consistency of community detection by reducing the impact of stochastic variations and ensures a more reliable partitioning of the network into meaningful communities.

Figures 10, 11, 12, 13, 14, and 15, depicts the community cluster using hierarchical clustering for modules network,$\mathbf{P}^M$.

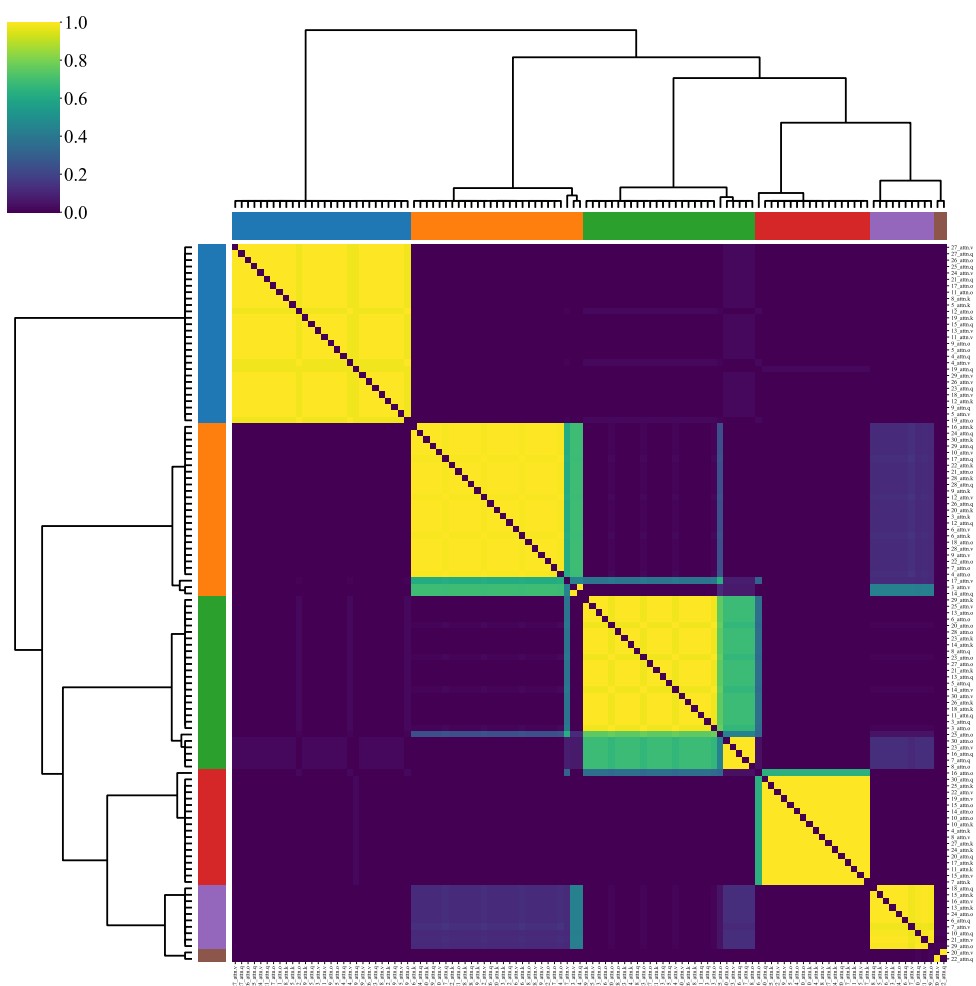

Figure 10: **Heat map clustering of modules network ($\mathbf{P}^M$) for the llama model with block-based pruning, where leaf colors in the dendrograms represent distinct communities formed through hierarchical clustering of the co-assignment matrix, revealing structural patterns among attention modules across layers.**

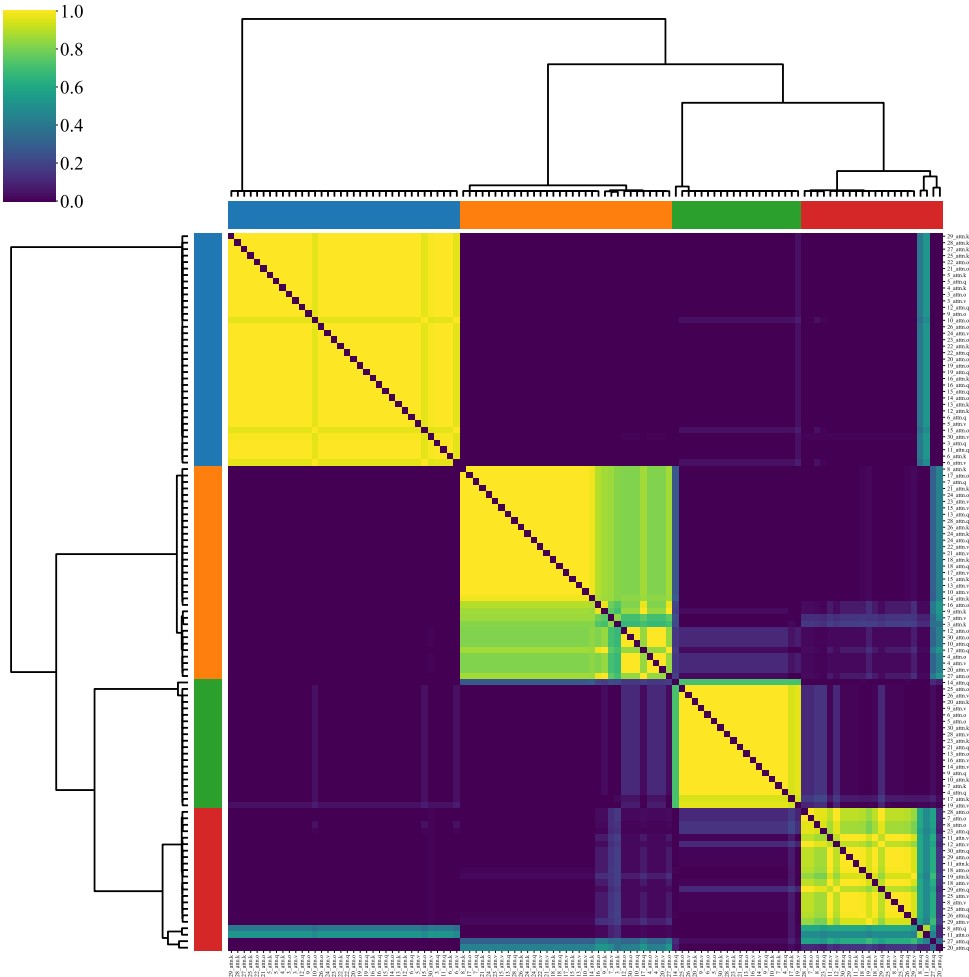

Figure 11: **Heat map clustering of modules network ($\mathbf{P}^M$) for the llama-chat model with block-based pruning, where leaf colors in the dendrograms represent distinct communities formed through hierarchical clustering of the co-assignment matrix, revealing structural patterns among attention modules across layers.**

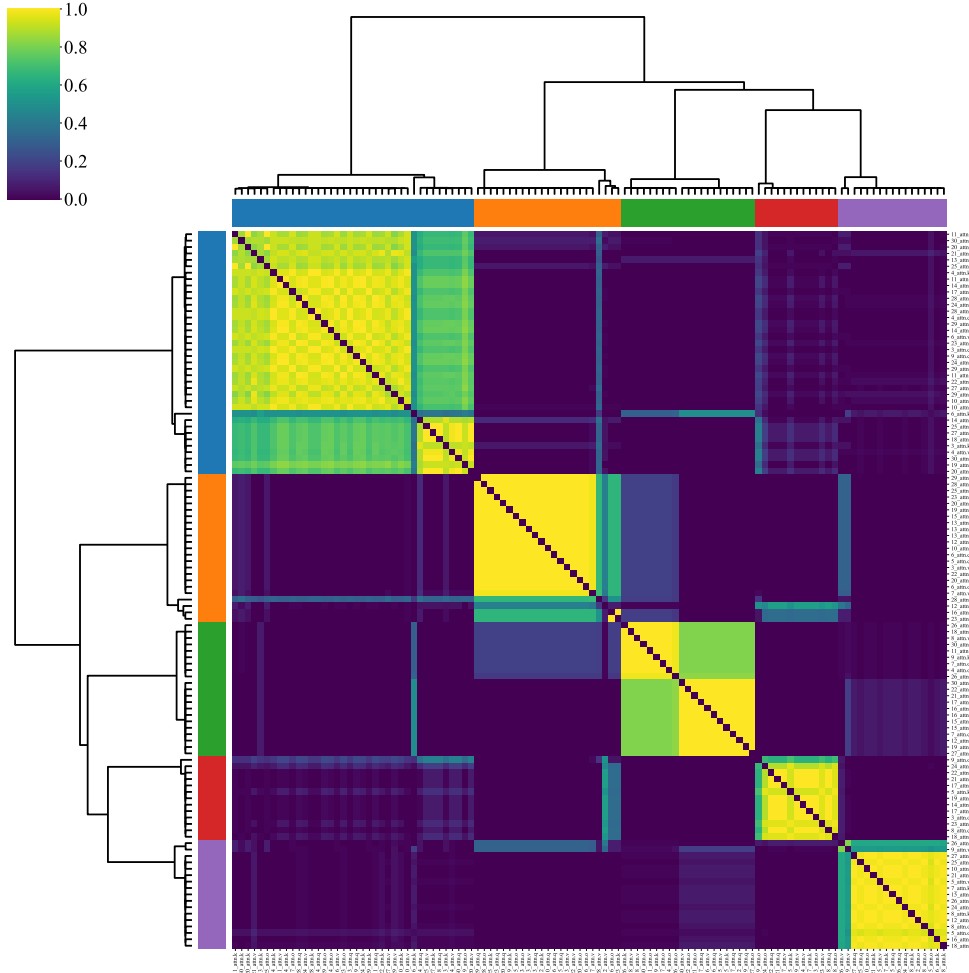

Figure 12: **Heat map clustering of modules network ($\mathbf{P}^M$) for the vicuna model with block-based pruning, where leaf colors in the dendrograms represent distinct communities formed through hierarchical clustering of the co-assignment matrix, revealing structural patterns among attention modules across layers.**

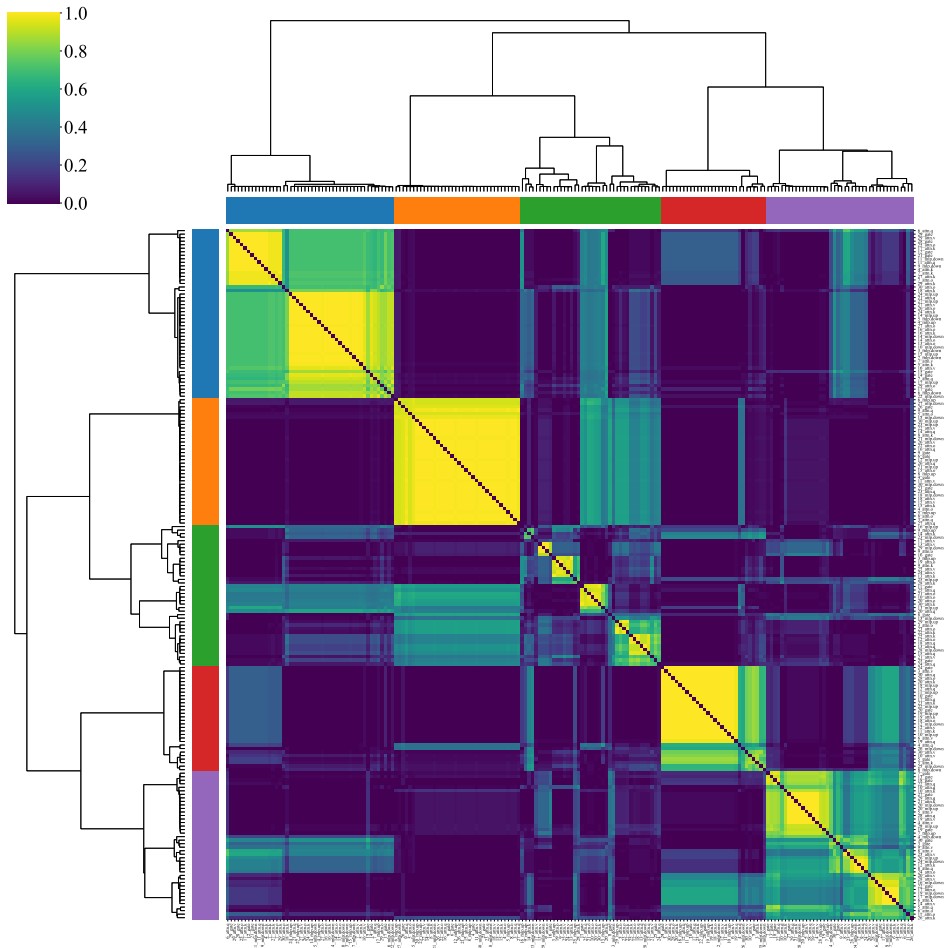

Figure 13: **Heat map clustering of modules network ($\mathbf{P}^M$) for the llama model with channel-based pruning, where leaf colors in the dendrograms represent distinct communities formed through hierarchical clustering of the co-assignment matrix, revealing structural patterns among attention modules across layers.**

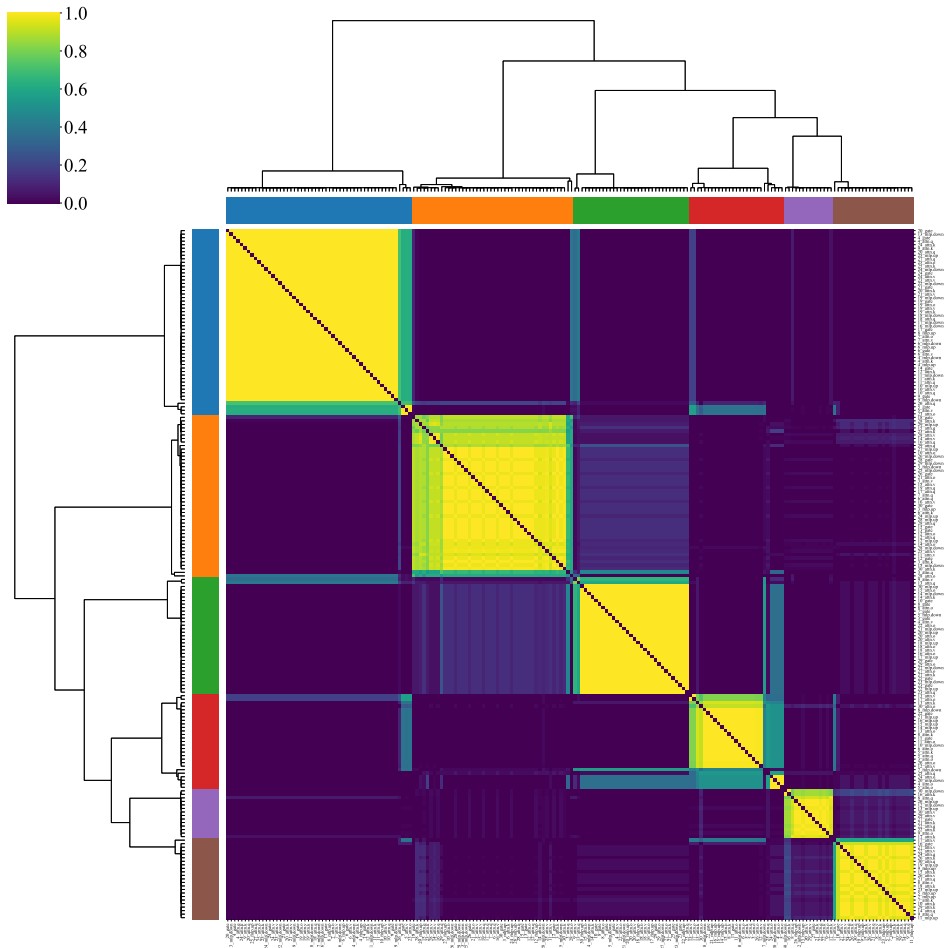

Figure 14: **Heat map clustering of modules network ($\mathbf{P}^M$) for the llama-chat model with channel-based pruning, where leaf colors in the dendrograms represent distinct communities formed through hierarchical clustering of the co-assignment matrix, revealing structural patterns among attention modules across layers.**

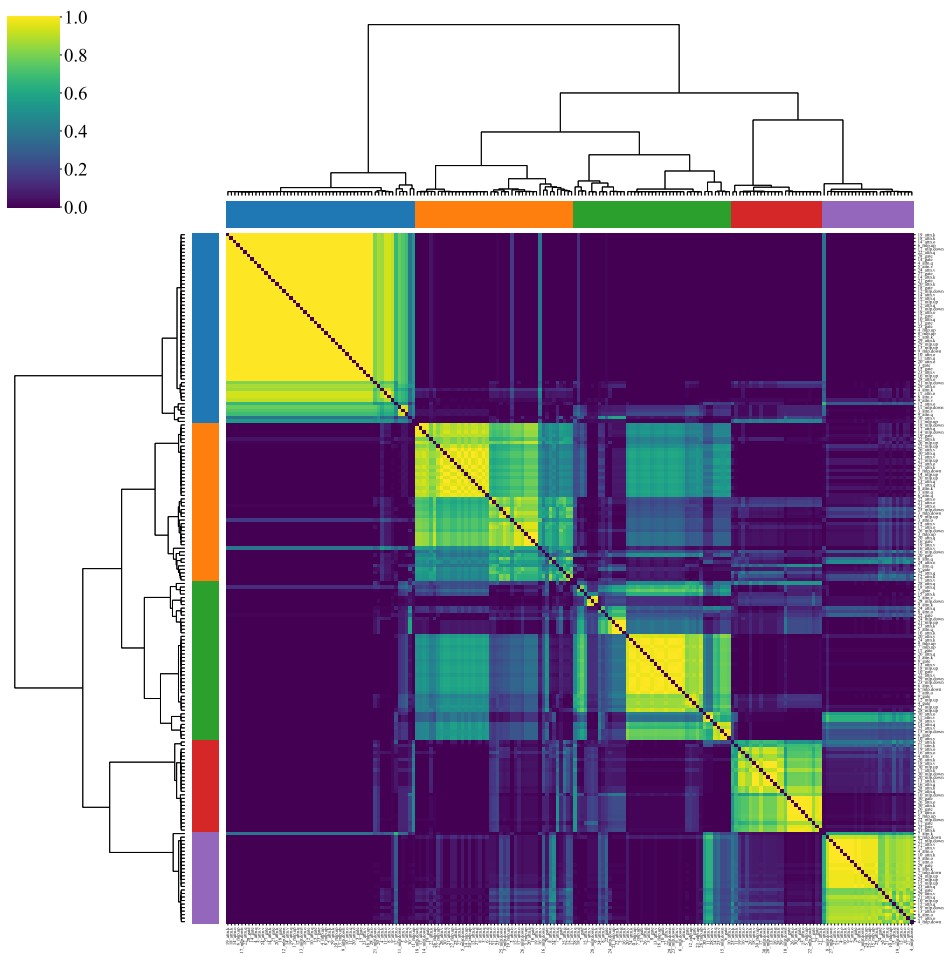

Figure 15: **Heat map clustering of modules network ($\mathbf{P}^M$) for the vicuna model with channel-based pruning, where leaf colors in the dendrograms represent distinct communities formed through hierarchical clustering of the co-assignment matrix, revealing structural patterns among attention modules across layers.**

## Supplementary Note 5: Emerging community structures in skill and module networks

The architecture of emerging LLMs reflects the self-organization of neural assemblies [56], where local activity and interactions drive the emergence of specific, stereotyped connectivity patterns, such as modularity. Constructing the overall network topology becomes essential. It involves integrating the dataset that captures domain knowledge, the architectural modules of LLMs, and the emerging cognitive functions. The methodology for building these networks is detailed in the Supplementary Information (SI 1 and SI 2). Network visualization reveals the structural and functional relationships among skills, datasets, and modules, and illustrates the process used to generate the Skills and Modules projection networks. These networks exhibit distinct connectivity patterns that can be leveraged to study the localization of skills within LLM modules, an essential step toward understanding the emergence of intelligence.

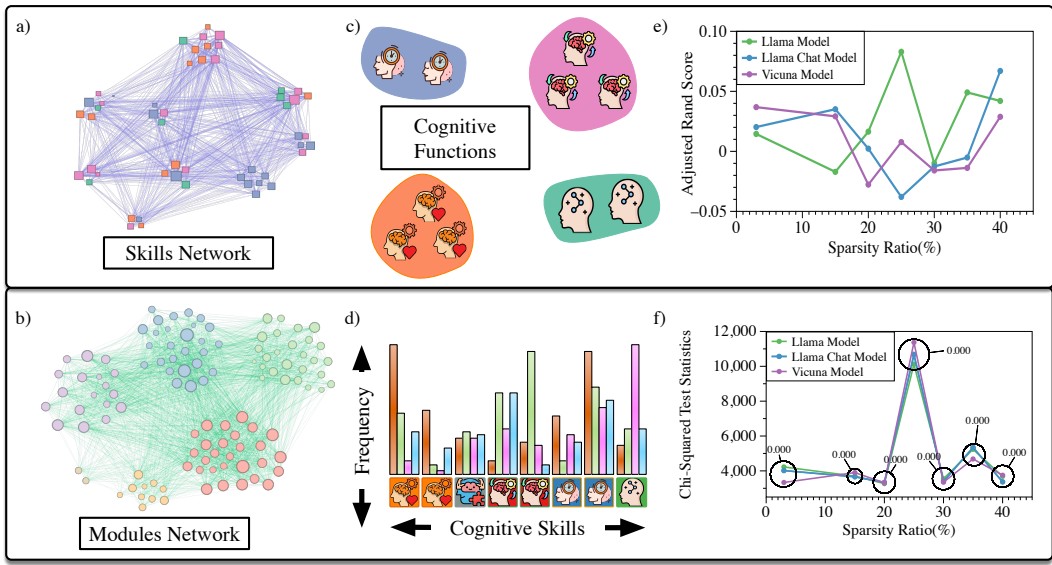

Figure 16: **Community structure comparison between skills and modules networks.** **(a)** The skills projection network with nodes (skills) grouped based on the Louvain community detection algorithm, and colored with the cognitive-function label taken from Table SI1, allowing direct visual comparison between detected communities and domain ground truth. **(b)** The Modules projection network, where each node (module) is assigned a community label based on Louvain partitioning applied at multiple resolutions, subsequently consolidated using average-linkage hierarchical clustering.**(c)** Color legend of cognitive function for node color in skills network in **(a)**. **(d)** Schematic figure to represent the frequency of cognitive skills within each community for the Modules network**(b)**. **(e)** Adjusted Rand Score (ARS) between the Louvain communities in the Skills network and the cognitive-function labels, plotted against different sparsity ratio for pruning and three base models (Llama, Llama-Chat, and Vicuna). **(f)** The chi-squared T-test statistically assessed the distinctiveness of skill distributions within each community of Modules networks, with their p-value for those three different models for different sparsity ratios.

Growing observations demonstrate that neural networks often exhibit meaningful community organization. Therefore, we leverage Louvain community detection techniques[9] to uncover latent interdependencies and organizational patterns among cognitive skills (Figure 16a) and LLM modules (Figure 16b). The resulting community structures reveal a hierarchical and modular architecture within LLMs, shedding light on how localized and distributed processing underpins their cognitive capabilities. This insight carries significant implications for model design, interpretability, and optimization. Surprisingly, while groups of LLM modules are tightly interconnected through shared skill distributions, there is no precise alignment between the predefined cognitive functions and the communities identified in the skills network (Figure 16c). A Chi-square test comparing the distribution of skills across communities is illustrated in Figure 16d, indicating that skill allocation is statistically independent of the predefined cognitive categories.

Next, we quantify the contribution of specific model components to performance across datasets under different pruning strategies and task distributions [44]. We use Adjusted Rand Index (ARI) scores to evaluate the normalized agreement between clusters by assessing all pairs of elements (see SI Section 4). Figure 16(**e**) shows the ARI scores for communities of skills and cognitive functions across various sparsity levels used in pruning the model [15]. Despite performance degradation with increased pruning or across different Llama2 model variants, ARI scores do not improve under any pruning strategy. This contrasts with the human brain, where specific skill types tend to localize within distinct cognitive regions [50, 11, 55, 12], suggesting that LLMs exhibit a different structural-functional organization. Across all pruning strategies, sparsity levels, and Llama2 model variants, p-values remained consistently and significantly low (<0.05), indicating that each community possesses a distinct skill distribution (Figure 16**f**). This implies that although specific skill types are not localized according to cognitive function the module localization still reflects unique combinations of skills.

## Supplementary Note 6: Influence of Modules with each community of Modules Network

### Within-Module Degree Z-Score

Degree Z-Score indicates how a module compares connectivity to others within the same community [29, 53]. A high Z-score means the module has more connections than typical for its community, suggesting a central or dominant role within that group.

The Z-Score of a module within its community:

$$Z_i = \frac{k_i - \mu_{C_i}}{\sigma_{C_i}}$$

Where:

- $Z_i$ is the Within-Module Degree Z-Score for module $i$.
- $k_i$ is the degree of module $i$.
- $\mu_{C_i}$ is the mean degree of the community $C_i$ to which module $i$ belongs.
- $\sigma_{C_i}$ is the standard deviation of the degrees within community $C_i$.

### Participation Coefficient

The Participation Coefficient is a measure used to quantify how a module is connected to multiple communities within the network[29, 53]. The Participation Coefficient for a module in a network is given by:

$$P_i = 1 - \sum_{s=1}^{n} \left( \frac{k_{is}}{k_i} \right)^2$$

Where:

- $P_i$ is the Participation Coefficient of module $i$.
- $k_{is}$ is the number of edges (or degree) that module $i$ has with nodes in community $s$.
- $k_i$ is the total degree (number of edges) of module $i$.
- The sum is taken over all $n$ communities in the network.

Figure 17, highlights the distribution of edge-weight as well as the the within-module degree Z-score analysis of different communities formed using different pruning strategies for different models. Th e within-module degree Z-score analysis highlights modules that exhibit significantly greater connectivity compared to other modules within the same community. A high Z-score identifies a module as central or dominant within its community, reflecting its critical role in facilitating internal communication and coherence. In parallel, the participation coefficient provides insights into the inter-community connectivity of modules, measuring the extent to which a module is interconnected across different communities. A higher participation coefficient indicates that a module bridges multiple communities, acting as an integrative or intermediary component within the broader

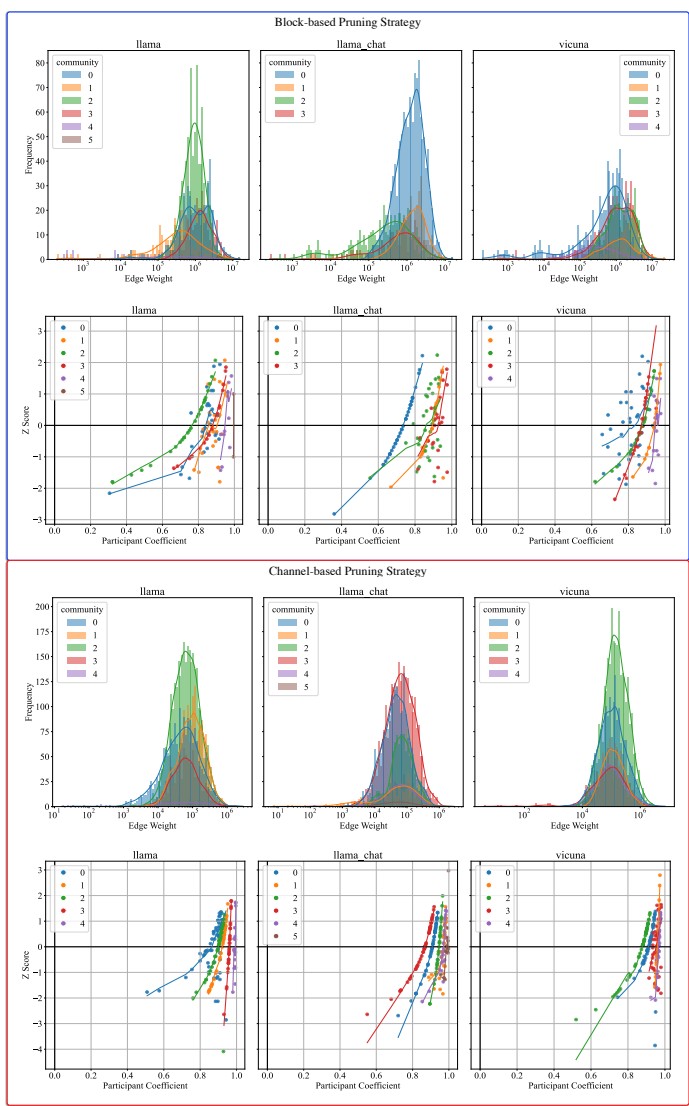

Figure 17: **Influence of Modules within each community of Modules Network.(a-b)** The distribution of edge weight within different communities of modules network for Block and Channel based modules.(**c-d**) The scatter plot for different modules over two metrics, participation coefficient and degree z-score metric.

network structure. Together, these metrics reveal nuanced roles of individual modules, distinguishing between community-specific hubs and those crucial for inter-community communication and network integration.

From the figure we see that, in both pruning strategies, most modules exhibit relatively high participation coefficients (typically between 0.6 and 1.0), suggesting a network where modules generally maintain connections across multiple communities rather than being strictly confined to their local communities. The within-module Z-scores display considerable variability across the modules, ranging approximately between -3 and +3 for block-based pruning and between -4 and +3 for channel-based pruning. Higher positive Z-scores (above zero) indicate modules functioning as local community hubs with stronger intra-community connections. Conversely, negative Z-scores suggest peripheral roles with fewer local connections.

The distinct clustering and spread patterns indicate that the block-based pruning strategy leads to a network with more defined module roles (either community-centric or integrative), enabling clearer interpretability of how skills might be localized within specific modules or communities. Conversely,

the channel-based pruning strategy yields networks with uniformly high cross-community integration, suggesting that this strategy may reduce clarity about functional specialization but highlights the distributed and interconnected nature of module interactions. These observations underscore the structural complexity in LLMs, where network modules exhibit diverse roles. Such roles likely influence how abstract cognitive skills are encoded and integrated throughout the model architecture, reflecting an interplay between local specialization and global integration.

## Finetuning Details

Drawing from the hypothesis that modules associated with distinct skill distributions play specialized roles, we aligned task datasets with corresponding module communities using KL divergence to capture the closest match between dataset and module specialization. Figure 18 illustrates the methodology for how communities based on a specific distribution of skills can be used to fine-tune the model based on their cognitive skill relevance.

We employ distributed training and evaluation to analyze the performance of LLMs, including Llama, Llama-chat, and Vicuna, fine-tuned using datasets aligned with cognitive skill-based module communities. Models are initialized with pre-trained weights, with specific modules (e.g., attn.q, map.up) selectively frozen or fine-tuned based on three strategies: community-specific, random, or all modules. We froze all the parameters not included when creating the communities. Randomized module subsets are generated by replacing community modules with non-community equivalents to evaluate robustness. Random modules closely relate to the community of modules, i.e., if an attn.q module is in the community, then the random subset contains attn.q module that is 1 or 2 layers different. Distributed training leverages NVIDIA's NCCL backend for inter-GPU communication, with AdamW as the optimizer and hyperparameters set to five epochs, a batch size of two, and a learning rate of 0.00001. Mixed precision (bfloat16) and Fully Sharded Data Parallel (FSDP) strategies, including CPU offloading, ensure computational efficiency and memory optimization. Skill-aligned datasets are used for fine-tuning, with a validation size of 100 samples and a top-skill selection strategy to match datasets to community skill profiles. Model evaluation computes accuracy by comparing predicted logits with true labels alongside metrics such as Euclidean magnitude of weight changes and L2 norm for weight sparsity. This integrated approach enables a detailed understanding of how cognitive skill alignment influences LLM performance.

### Performance on Targeted Finetuning

Figure 19 and 20 show the impact of targeted finetuning using the community of modules as depicted in figure 18. The results demonstrate crucial insights into the comparative effects of targeted finetuning using communities formed through two pruning strategies—block-wise and channel-wise—on the performance and structural adaptation of fine-tuned LLMs. In both pruning conditions, fine-tuning across all modules consistently achieved the highest accuracy for all models tested (Llama, Llama-Chat, and Vicuna), clearly surpassing both community-based and random-module fine-tuning. Intriguingly, the accuracy obtained through fine-tuning community-based modules, selected based on cognitive skill associations, did not significantly differ from that achieved by randomly selected modules under either pruning strategy. This result underscores that the assumed specialization of LLM modules tied explicitly to cognitive functions does not translate into enhanced performance relative to random module selection, irrespective of pruning strategy.

The L2 norm differences in weight updates reveal nuanced distinctions between the two pruning methods. Since we fixed the hyperparameter to be the same for finetuning, the learning rate remains the same. Hence, the magnitude difference represents the gradient norm. Under both block-wise and channel-wise pruning, community-based fine-tuning led to notably more significant magnitude change compared to all-module or random-module fine-tuning. This suggests that community-based fine-tuning is more sensitive to fine-tuning than other finetuning. Nevertheless, despite the sensitivity, community-based fine-tuning did not yield proportional improvements in accuracy over random selections, an observation consistent across both pruning approaches. Moreover, the magnitude of weight updates under block-wise pruning generally exceeded that observed in channel-wise pruning, suggesting that block-wise pruning induces more pronounced structural modifications within targeted modules. Figures 21, 22, and 23 illustrated individual magnitude differences of each module within the community to the original pre-trained modules for different models and pruning strategies.

Collectively, these findings highlight two important insights: first, the limited efficacy of predefined cognitive-skill module selection for enhancing fine-tuning performance remains consistent across different pruning strategies; second, block-wise pruning triggers more substantial structural updates than channel-wise pruning, yet this greater magnitude of change does not translate into superior accuracy gains. These results reinforce the broader conclusion that LLMs encode knowledge through distributed rather than strictly modular specializations.

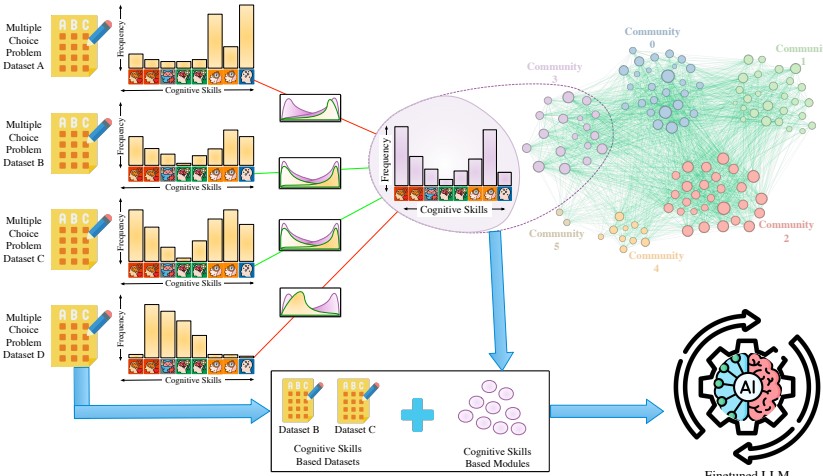

Figure 18: **Community-based fine-tuning aligned with cognitive skill relevance** The influence of skill distributions within identified module communities is examined by selecting datasets matching the skill profiles of these communities.

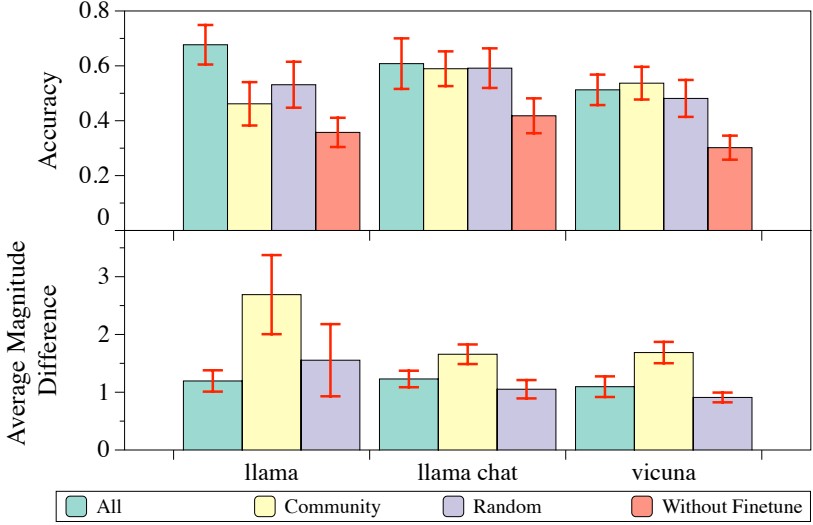

Figure 19: **Performance of Targeted Finetuning** Accuracy and weight difference magnitude of fine-tuned models (Llama, Llama-Chat, Vicuna) across two datasets aligned with each community that were created using **block**-based pruning strategy.

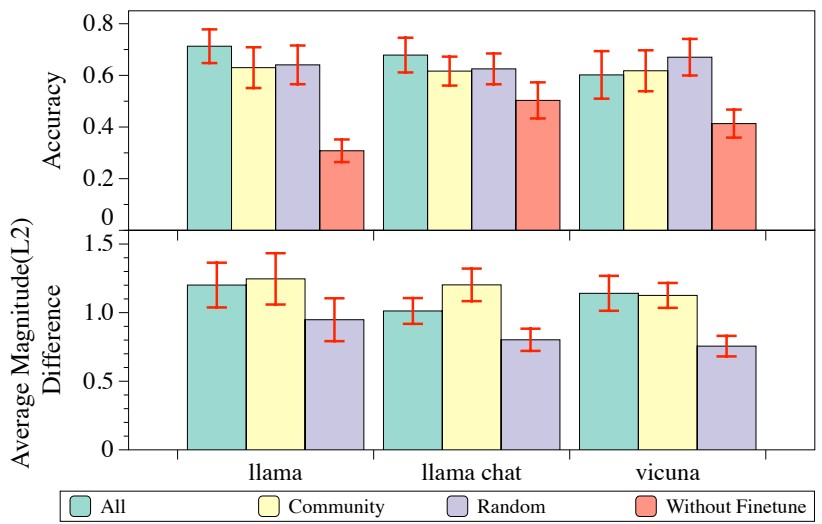

Figure 20: **Performance of Targeted Finetuning** Accuracy and weight difference magnitude of fine-tuned models (Llama, Llama-Chat, Vicuna) across two datasets aligned with each community that were created using **channel**-based pruning strategy.

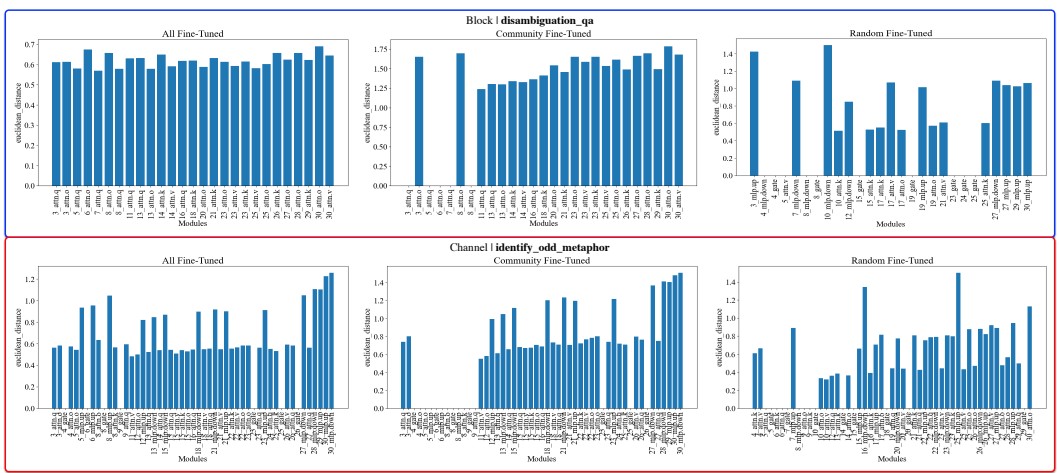

Figure 21: **Visualization of changes in weight modules of the Llama model after fine-tuning, highlighting task associations such as 'disambiguation_qa' (Block) and 'identify_odd_metaphor' (Channel).**

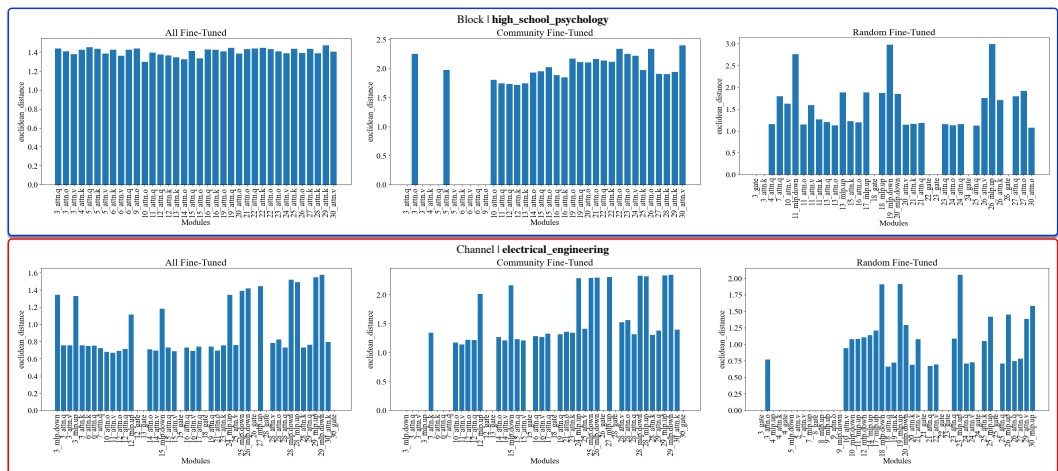

Figure 22: **Visualization of changes in weight modules of the Llama-Chat model after fine-tuning, highlighting task associations such as 'high_school_psychology' (Block) and 'electrical_engineering' (Channel).**

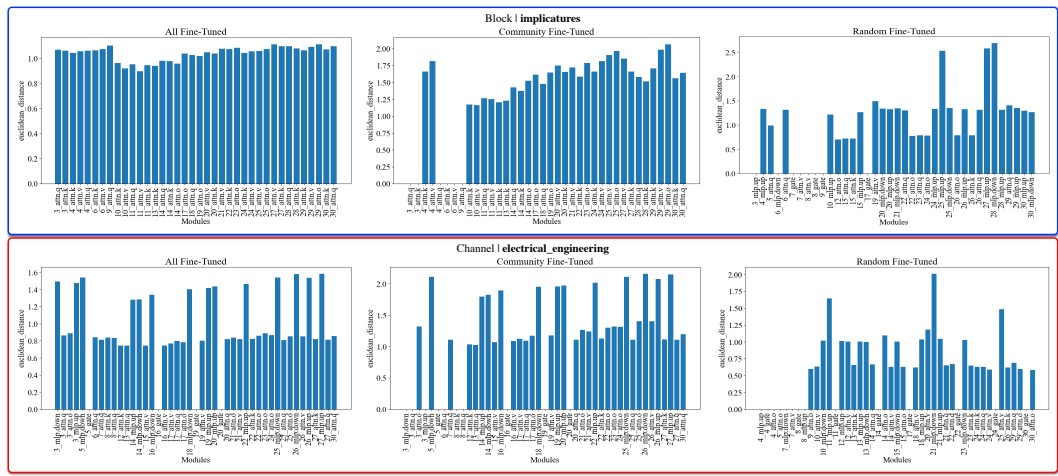

Figure 23: **Visualization of changes in weight modules of the Vicuna model after fine-tuning, highlighting task associations such as 'implicatures' (Block) and 'electrical_engineering' (Channel).**

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
