# OpenReview forum: "Unraveling the cognitive patterns of Large Language Models through module communities"
_NeurIPS.cc/2025/Workshop/UniReps — UniReps2025_

### Official Review · Reviewer_zbZy · 2025-09-09
**Unraveling the cognitive patterns of Large Language Models through module communities**

**Confidence:** 2

**Review:**

This paper proposes a framework for interpreting large language models (LLMs) by analyzing their internal modular structure through the lens of biological cognition. The authors draw parallels between the modular architecture of biological brains and the organization of LLMs, using network science methods to explore how cognitive skills such as memory, attention, and language processing are represented within model components.
The analysis focuses on models including LLaMA, LLaMA Chat, and Vicuna. The authors examine the eigenvalue spectra, participation coefficients, and Z-scores of module networks to characterize their structure. These metrics reveal that while LLM modules form densely connected communities, they also maintain weaker links across communities, resembling the ``weak-localization'' architectures seen in birds and small mammals rather than the strongly modular organization of the human brain.
The paper further evaluates the impact of different fine-tuning strategies. It compares random, all-module, and community-specific module fine-tuning using datasets aligned with specific cognitive skills. While community-based fine-tuning leads to the largest weight changes, full-module fine-tuning achieves the highest performance. This suggests that LLMs encode knowledge in a distributed, redundant fashion rather than in strictly localized modules.

Weaknesses:
- Missing Methodological Details for an Extended Abstract:
-> The abstract reads more like the introduction and framing of a full-length journal article than a focused NeurIPS extended abstract. While the conceptual setup is compelling and well-motivated, it occupies a LOT of real estate. As a result, critical methodological components—such as how the skill-to-dataset mappings are defined, how pruning was implemented, or how fine-tuning configurations were executed—are either briefly stated or omitted entirely.
--> I suggest  to at leas include one paragraph per method (community detection, dataset alignment, fine-tuning setup) would significantly improve the standalone readability of the abstract. This brings me to my next point:

- Lack of clarity of key terms used in the main abstract (e.g., what exactly are module communities?)
Given that this is a 4-page extended abstract, it is understandable that not all methods can be detailed. However, the dependency on the 26-page supplement is too strong—particularly for core definitions like “community” or “skill mapping.” A reader skimming the abstract without the supplement would be left without a functional understanding of the methods.



--> Suggestion: Provide inline references with HYPERLINKS to key sections of the supplement (e.g., "see  Sec. 2 for community detection") and ensure that every foundational term is at least minimally explained in the abstract.

- Figure Presentation and Captioning Issues
The heatmaps in the supplement  could be improved for clarity. In particular:
- The modules in the heatmaps are unlabeled, making it hard to interpret the alignment between skills, datasets, and communities.
- Captions across similar subfigures are nearly identical, making it difficult to quickly identify key differences when the entire caption is bolded throughout the heat map plots

-->Suggestion: Label axes and module indices directly in the heatmaps. Use bold text only for the unique aspects of each subcaption rather than the full text.

Strengths:
- Introduces a novel interdisciplinary approach integrating neuroscience and network theory into LLM interpretability.
- Provides insight into how LLMs distribute cognitive functions across network modules.

My final, general advice would be: make it easy for the reviewer. Don’t make me jump to the supplements when I am missing clarity of terms, and if you send me to the abstract, use hyperlinks!  I am intrigued by the interdisciplinary approach this paper proposes and I think this work has a lot of potential but I think it has a lot of room of improvement to bring across the most relevant pieces  within the 4 page constraint. I honestly didn’t have the time to closely read the abstract which is why I am assigning low scores.

**Score:**

3

**Topic Fit:**

3

---

### Official Review · Reviewer_soXe · 2025-09-10
**Review of Unraveling the cognitive patterns of Large Language Models through module communities**

**Confidence:** 4

**Review:**

This paper constructs a multipartite network linking cognitive skills, LLM architectures, and datasets, then analyze the modular structure using network metrics (eigenvalue distribution, participation coefficient, Z-score).

Strengths
- The integration of network science, cognitive science, and machine learning is potentially impactful.
- The categorization of 50+ cognitive skills across four domains (cognitive process memory, executive function, language communication, social cognition) is well-motivated.
-The use of spectral properties, participation coefficients, and Z-scores provides clear quantitative measures for characterizing modular structure across multiple LLM architectures.

Weaknesses
- Only three models tested (Llama, Llama Chat, Vicuna), all from the same family.
- No comparison to existing LLM interpretability methods (probing, attention visualization, etc.) to contextualize the approach's advantages.
- "Module" is used to refer to both network communities and architectural components, creating confusion.
- Figure 2 is difficult to interpret - the community colors/labels are unclear, and the relationship between panels (a-c) and (d-f) needs better explanation.
- The spectral analysis lacks theoretical justification for why eigenvalue distributions should reveal meaningful modular structure in this context.

Questions
- How do results change with different community detection algorithms? Have you tested robustness to network construction choices?
- The biological analogies are compelling but lack quantitative validation. Can you provide metrics comparing LLM network properties to actual neural connectivity data?
- Beyond fine-tuning insights, what other applications does this framework enable (e.g., model compression, architecture design)?

**Score:**

3

**Topic Fit:**

2

---

### Official Review · Reviewer_WbcL · 2025-09-15
**Interesting Investigation of Modularity in LLMs**

**Confidence:** 3

**Review:**

The paper draws on literature in neuroscience and AI to investigate the concept of modularity in large language models (LLMs).

Pros:

1. The work is a novel contribution in understanding how modularity works in LLMs and what benefits they bring to LLM finetuning.
2. There is relevant grounding in literature and it engages well with related works.
3. The motivation and methodology is clearly explained along with relevant experimental results.

Cons:

1. The main paper is lacking in details and most relevant technical information is present in the appendix. There should be more references to relevant section of Appendix in the main paper to enhance readability.

The work presents an interesting approach to a new topic of potential relevance- modular behaviour of cognitive systems!

**Score:**

4

**Topic Fit:**

3

---

### Official Review · Reviewer_kRc6 · 2025-09-16
**Study of LLM mechanisms through module detection in phenomenological networks**

**Confidence:** 4

**Review:**

This project seeks to understand the mechanistic structure of LLMs, and their architecture, through module detection of phenomenological networks constructed with LLM analysis of multiple-choice questions in datasets. While the overall question is interesting and ambitious, the implementation seemed underwhelming to me insofar as the skill-datasets networks capture phenomenological relationships at best, and there is a wide gap between the studied high-level interactions and the basic underpinning mechanisms of LLMs that the authors seek to characterize. It was unclear to me why the discovery of modules in these networks is surprising (i.e., unexpected in a statistical sense) and how this discovery can help, in principle, advance our understanding of the functional mechanisms by which LLMs work.

**Score:**

2

**Topic Fit:**

2